# Growth rules for the repair of Asynchronous Irregular neuronal networks after peripheral lesions

**Ankur Sinha**[1]*, **Christoph Metzner**[1,2], **Neil Davey**[1], **Roderick Adams**[1],
**Michael Schmuker**[1], **Volker Steuber**[1]

**1** UH Biocomputation Research Group, Centre for Computer Science and Informatics Research, University of Hertfordshire, Hatfield United Kingdom, **2** Department of Software Engineering and Theoretical Computer Science, Technische Universität Berlin, Berlin, Germany

* a.sinha2@herts.ac.uk

**Data Availability Statement:** The complete source code of all simulations run in this work are available on GitHub at https://github.com/sanjayankur31/SinhaEtAl2021 and also archived on

## Abstract

Several homeostatic mechanisms enable the brain to maintain desired levels of neuronal activity. One of these, homeostatic structural plasticity, has been reported to restore activity in networks disrupted by peripheral lesions by altering their neuronal connectivity. While multiple lesion experiments have studied the changes in neurite morphology that underlie modifications of synapses in these networks, the underlying mechanisms that drive these changes are yet to be explained. Evidence suggests that neuronal activity modulates neurite morphology and may stimulate neurites to selective sprout or retract to restore network activity levels. We developed a new spiking network model of peripheral lesioning and accurately reproduced the characteristics of network repair after deafferentation that are reported in experiments to study the activity dependent growth regimes of neurites. To ensure that our simulations closely resemble the behaviour of networks in the brain, we model deafferentation in a biologically realistic balanced network model that exhibits low frequency Asynchronous Irregular (AI) activity as observed in cerebral cortex. Our simulation results indicate that the re-establishment of activity in neurons both within and outside the deprived region, the Lesion Projection Zone (LPZ), requires opposite activity dependent growth rules for excitatory and inhibitory post-synaptic elements. Analysis of these growth regimes indicates that they also contribute to the maintenance of activity levels in individual neurons. Furthermore, in our model, the directional formation of synapses that is observed in experiments requires that pre-synaptic excitatory and inhibitory elements also follow opposite growth rules. Lastly, we observe that our proposed structural plasticity growth rules and the inhibitory synaptic plasticity mechanism that also balances our AI network both contribute to the restoration of the network to pre-deafferentation stable activity levels.

## Author summary

An accumulating body of evidence suggests that our brain can compensate for peripheral lesions by adaptive rewiring of its neuronal circuitry. The underlying process, structural

ModelDB at http://modeldb.yale.edu/267035. The scripts used to analyse the data generated by the simulation are available in a separate GitHub repository at https://github.com/sanjayankur31/Sinha2016-scripts. Post-processed data used to generate the figures in the paper, along with the plotting scripts are available at https://github.com/sanjayankur31/SinhaEtAl2021-figure-scripts. The raw data generated by the simulations used in this paper is available on Zenodo at https://zenodo.org/record/4727700/. These source code repositories are licensed under the Gnu GPL license (version 3 or later), and the data on Zenodo is licensed under a Creative Commons Attribution 4.0 International license.

**Funding:** The author(s) received no specific funding for this work.

**Competing interests:** The authors have declared that no competing interests exist.

plasticity, can modify the connectivity of neuronal networks in the brain, thus affecting their function. To better understand the mechanisms of structural plasticity in the brain, we have developed a novel model of peripheral lesions and the resulting activity-dependent rewiring in a simplified balanced cortical network model that exhibits biologically realistic Asynchronous Irregular (AI) activity. In order to accurately reproduce the directionality and course of network rewiring after injury that is observed in peripheral lesion experiments, we derive activity dependent growth rules for different synaptic elements: dendritic and axonal contacts. Our simulation results suggest that excitatory and inhibitory synaptic elements have to react to changes in neuronal activity in opposite ways. We show that these rules result in a homeostatic stabilisation of activity in individual neurons. In our simulations, both synaptic and structural plasticity mechanisms contribute to network repair. Furthermore, our simulations indicate that while activity is restored in neurons deprived by the peripheral lesion, the temporal firing characteristics of the network may not be retained by the rewiring process.

## Introduction

Multiple plasticity mechanisms act simultaneously and at differing time scales on neuronal networks in the brain. Whilst synaptic plasticity is limited to the changes in efficacy of pre-existing synapses, *structural* plasticity includes the formation and removal of whole neurites and synapses. Thus, structural plasticity can cause major changes in network function through alterations in connectivity. Along with confirmation of structural plasticity in the adult brain [1–4], recent work has also shown that axonal boutons and branches [5–10], and both inhibitory [11, 12] and excitatory dendritic structures [13, 14] are highly dynamic even in physiological networks.

Stability in spite of such continuous plasticity requires homeostatic forms of structural plasticity. A multitude of lesion experiments provide evidence for homeostatic structural plasticity [15–26]. A common feature observed in these studies is the substantial network reorganisation that follows deafferentation. Recent time-lapse imaging studies of neurites in the cortex during the rewiring process show that both axonal [6, 10, 27] and dendritic structures display increased turnover rates [10, 13, 28, 29] in and around the area deafferented by the peripheral lesion, the Lesion Projection Zone (LPZ). Specifically, while excitatory neurons outside the LPZ sprout new axonal collaterals into the LPZ, inhibitory neurons inside the LPZ extend new axons outwards [6]. Along with an increased excitatory dendritic spine gain [28] and a marked loss of inhibitory shaft synapses [11, 30] in the LPZ, the rewiring of synapses in the network successfully restores activity to deprived LPZ neurons in many cases.

Access to such data and recent advances in simulation technology have enabled computational modelling of activity dependent structural plasticity [31–38]. In their seminal work, Butz and van Ooyen introduced the Model of Structural Plasticity (MSP) framework [31]. They demonstrated its utility by simulating a peripheral lesioning study to explore the activity dependent growth rules of neurites [33, 34]. Their analysis suggests that the restoration of activity could only be caused by the experimentally noted inward increase in excitatory lateral projections into the LPZ when dendritic elements sprouted at a lower level of activity than their axonal counterparts. Further, since excitatory and inhibitory synaptic elements were treated identically in their model and this results in inhibitory projections also flowing into the LPZ instead of growing outwards from the LPZ, Butz and van Ooyen also discuss that the contribution of inhibitory neurons to the repair process remains an important open question. A

computational model of peripheral lesioning that reproduces all features of the repair process in cortical networks is therefore still lacking.

Here, as the next step towards improving our understanding of activity dependent structural plasticity in cortical networks, we build on Butz and van Ooyen's work to re-investigate activity dependent growth rules for neurites in the biologically plausible cortical network model developed by Vogels, Sprekeler et al. [39]. Unlike in [33] where the cortical network to be deafferented was "grown" using a pre-set free parameter, the Vogels, Sprekeler network model explicitly incorporates cortical network characteristics. Additionally, it is also balanced by homeostatic inhibitory Spike Timing Dependent Plasticity (STDP) to a low frequency Asynchronous Irregular (AI) (spontaneous) firing regime as observed in the mammalian cortex [40, 41] and has been demonstrated to function as an attractor-less store for associative memories [39]. By deafferenting this network and reproducing the course of repair as reported in experimental work, we systematically derive activity dependent growth rules for all neurites—excitatory and inhibitory, pre-synaptic and post-synaptic.

Our simulations show that not all neurons in the inhibition-balanced Vogels, Sprekeler cortical network experience a loss in activity after deafferentation. Whereas neurons in the LPZ lose activity after deafferentation, neurons outside the LPZ gain extra activity—due to a net loss in inhibition. As a result, the growth rules proposed by Butz and van Ooyen for inhibitory neurites, in which neurites only sprout when neurons lose activity, do not result in repair here. Instead, our simulations suggest that excitatory and inhibitory neurites follow opposite activity dependent growth rules. We show that these new growth rules correctly simulate the ingrowth of excitatory projections into and the outgrowth of inhibitory projections from the deafferented area. Although deduced from network simulations, we also find that the post-synaptic growth rules contribute to the maintenance of activity in individual neurons by re-establishing their balance between excitation and inhibition (E-I balance). Furthermore, we observe that both homeostatic processes in our model—synaptic plasticity and structural plasticity—contribute to the repair process to successfully restore activity levels in neurons to pre-deafferentation levels. Our novel computational model of peripheral lesioning and repair makes important predictions about the activity dependent growth of neurites in balanced cortical networks, and provides a new platform to study the structural and functional consequences of peripheral lesions.

## Results

### A new model of recovery in simplified cortical AI networks after peripheral lesions

Our network model is based on the inhibition-balanced cortical network model developed by Vogels, Sprekeler et al. [39]. It consists of excitatory (E) and inhibitory (I) conductance based single compartment point neuron populations that are distributed in a toroidal grid and sparsely connected via exponential synapses [42]. Apart from inhibitory synapses projecting from the inhibitory neurons to the excitatory ones (IE synapses), whose weights are modified by Vogels, Sprekeler symmetric inhibitory STDP, all synaptic conductances (II, EI, EE) are static (Fig 1A). Structural plasticity when enabled, however, acts on all synapses in the network. In its steady state the STDP mechanism maintains the E-I balance in the network which then exhibits low frequency spontaneous Asynchronous Irregular (AI) firing characteristics (see Methods).

We simulate a peripheral lesion in the network by deafferenting a spatial selection of neurons to form the LPZ. For analysis, following experimental lesion studies, we divide the neuronal population into four regions relative to the LPZ (Fig 1B). The LPZ is divided into two regions:

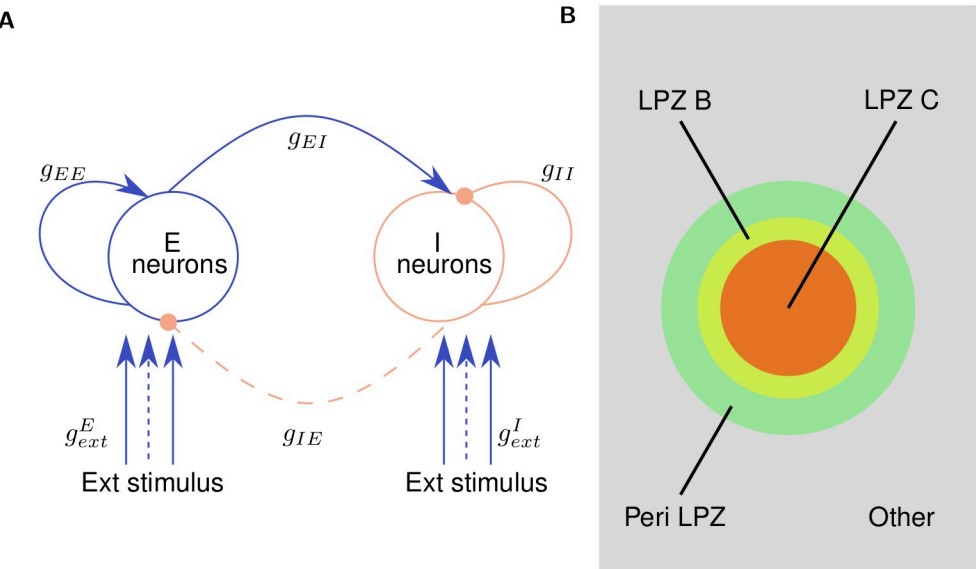

**Fig 1. Overview of the model.** (A) Excitatory (E) and Inhibitory (I) neurons ($N_E = 4N_I$ (see Methods)) are initially connected via synapses with a connection probability of ($p = 0.02$). All synapses (EE, EI, II), other than IE synapses, which are modulated by inhibitory spike-timing dependent plasticity, are static with conductances $g_{EE}$, $g_{EI}$, $g_{II}$, respectively. All synapse sets are modifiable by the structural plasticity mechanism. External Poisson spike stimuli are provided to all excitatory and inhibitory neurons via static synapses with conductances $g_{ext}^E$ and $g_{Inh}^I$, respectively. To simulate deafferentation, the subset of these synapses that project onto neurons in the Lesion Projection Zone (LPZ) (represented by dashed lines in the figure) are disconnected. (B) Spatial classification of neurons in relation to the LPZ: LPZ C (centre of LPZ) consists of 2.5% of the neuronal population; LPZ B (inner border of LPZ) consists of 2.5% of the neuronal population; Peri-LPZ (outer border of LPZ) consists of 5% of the neuronal population; Other neurons consist of the remaining 90% of the neuronal population. (Figure not to scale).

- LPZ C: the centre of the LPZ (Red in Fig 1B).

- LPZ B: the inner border of the LPZ (Yellow in Fig 1B).

    Neurons outside the LPZ are further divided into two regions:

- P LPZ: peri-LPZ, the outer border of the LPZ (Green in Fig 1B).

- Other neurons: neurons further away from the LPZ (Grey in Fig 1B).

Following the MSP framework, each neuron possesses sets of both pre-synaptic (axonal) and post-synaptic (dendritic) synaptic elements, total numbers of which at each neuron are represented by ($z_{pre}$) and ($z_{post}$), respectively. Excitatory and inhibitory neurons only possess excitatory ($z_{pre}^E$) and inhibitory axonal elements ($z_{pre}^I$), respectively, but they can each host both excitatory and inhibitory dendritic elements ($z_{post}^E$, $z_{post}^I$) (Fig 2A). The rate of change of each type of synaptic element per simulation time step, ($dz/dt$), is modelled as a Gaussian function of the neuron's time averaged activity, referred to as its "calcium concentration" ($[Ca^{2+}]$):

$$
\begin{aligned}
\frac{dz}{dt} &= v\left(2\,\exp^{-\left(\frac{[Ca^{2+}]-\xi}{\zeta}\right)^2} - \omega\right)\\
\xi &= \frac{\eta + \epsilon}{2},\\
\zeta &= \frac{\eta - \epsilon}{2\sqrt{-\ln(\omega/2)}}
\end{aligned}
\tag{1}
$$

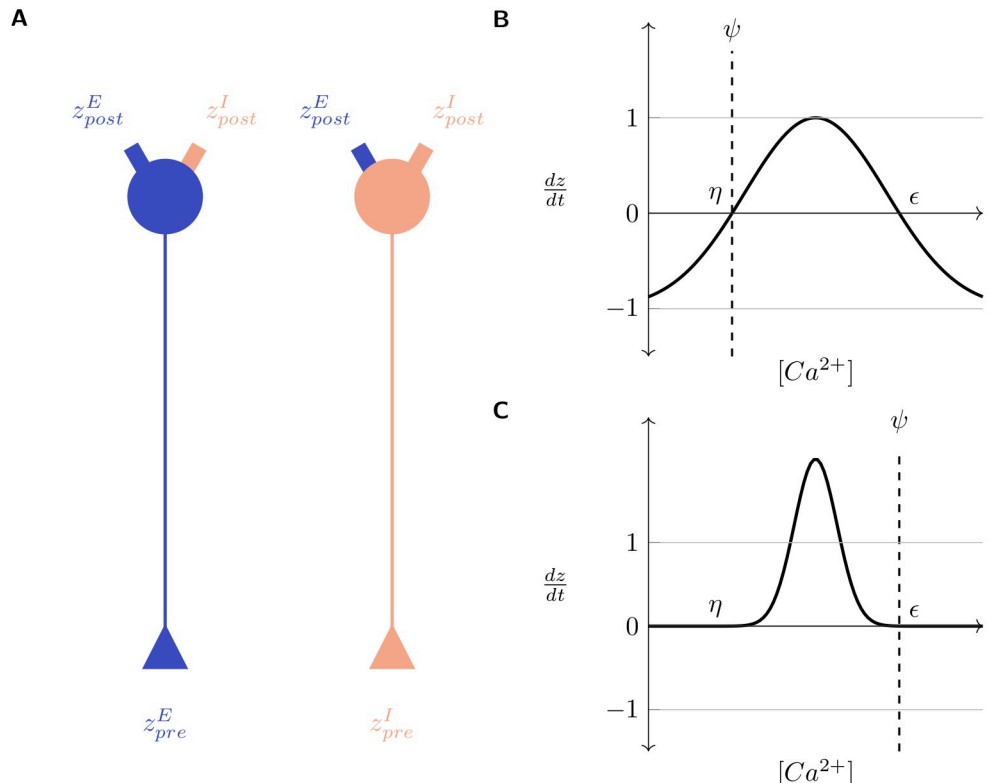

**Fig 2. Gaussian growth curves modulate the rate of turnover of synaptic elements ($\frac{dz}{dt}$) in a neuron as a function of its [$Ca^{2+}$].** (**A**) Excitatory: Blue; Inhibitory: Red; All neurons possess excitatory and inhibitory post-synaptic elements ($z^E_{post}, z^I_{post}$) but excitatory and inhibitory neurons can only bear excitatory and inhibitory pre-synaptic elements, respectively ($z^E_{pre}, z^I_{pre}$); (**B**) and (**C**). Example Gaussian growth curves. Constants $\eta$ and $\epsilon$ control the width and positioning of the growth curve on the x-axis. $\omega$ (see Eq 1) controls the positioning of the growth curve on the y-axis. $v$ (see Eq 1) is a scaling factor. $\psi$ is the optimal [$Ca^{2+}$] for the neuron. The minimum and maximum values of $dz/dt$ can be analytically deduced to be $-v\omega$ and $v(2 - \omega)$ respectively (see Methods). The relationship between $\eta$, $\epsilon$, and $\psi$ regulates the activity dependent dynamics of neurites. (**B**) $\psi = \eta = 5.0$, $\epsilon = 15.0$, $v = 1.0$, $v = 1.0$, $-v\omega = -1.0$, $v(2 - \omega) = 1.0$ (see Methods). Here, new neurites are formed when the neuronal activity exceeds the required level and removed when it falls below it. (**C**) $\eta = 5.0$, $\psi = \epsilon = 15.0$, $v = 1.0$, $\omega = 0.001$, $-v\omega = -0.001$, $v(2 - \omega) = 1.999$ (see Methods). Here, the growth curve is shifted up along the y-axis by decreasing the value of $\omega$. New neurites are formed when the neuronal activity is less than the homeostatic level and removed (at a very low rate) when it exceeds it.

Here, $v$ is a scaling factor and $\eta$, $\epsilon$ define the width and location of the Gaussian curve on the x-axis. Extending the original MSP framework, we add a new parameter $\omega$ that allows the translation of the growth curve along the y-axis (($\omega = 1$) returns the growth curves to the form included in the MSP). The relationship between $\eta$, $\epsilon$ and the optimal activity level of a neuron, $\psi$, govern the activity-dependent dynamics of each type of synaptic element. When its activity is at the optimal level ([$Ca^{2+}$] = $\psi$), a neuron is in its balanced steady state and should not turn over neurites. This implies that the growth curves must be placed such that $dz/dt = 0$ when [$Ca^{2+}$] = $\psi$. Hence, $\psi$ can take one of two values: ($\psi \in \{\eta, \epsilon\}$), and the turnover of synaptic elements dz/dt is:

$$
\begin{aligned}
&> 0 &&\text{for} \quad \eta < [Ca^{2+}] < \epsilon \\
&= 0 &&\text{for} \quad [Ca^{2+}] = \{\eta, \epsilon\} \\
&< 0 &&\text{for} \quad [Ca^{2+}] < \eta \quad \cup \quad [Ca^{2+}] > \epsilon
\end{aligned}
\tag{2}
$$

This is illustrated in Fig 2. Other than in a window between $\eta$ and $\epsilon$ where new neurites sprout, they retract. The new parameter $\omega$ allows the further adjustment of the growth curves to modulate the speeds of sprouting and retraction (Fig 2B and 2C). In Fig 2B with ($\psi = \eta$), new neurites will only be formed when the neuron experiences activity that is greater than its homeostatic value ($\psi < [Ca^{2+}] < \epsilon$). Fig 2C, on the other hand, shows the case for ($\psi = \epsilon$), where growth occurs when neuronal activity is less than optimal ($\eta < [Ca^{2+}] < \psi$).

The $[Ca^{2+}]$ for each neuron, a time averaged measure of its electrical activity, is calculated as:

$$\frac{d[Ca^{2+}]}{dt} = \begin{cases} -\frac{[Ca^{2+}]}{\tau_{[Ca^{2+}]}} + \beta, & \text{if } V \geq V_{th} \\[2ex] -\frac{[Ca^{2+}]}{\tau_{[Ca^{2+}]}}, & \text{otherwise.} \end{cases} \tag{3}$$

Here, $\tau_{[Ca^{2+}]}$ is the time constant with which $[Ca^{2+}]$ decays in the absence of a spike, $\beta$ is the constant increase in $[Ca^{2+}]$ caused by each spike, $V$ is the membrane potential of the neuron, and $V_{th}$ is the threshold membrane potential.

Figs 3 and 4 provide an overview of the activity in the network observed in an example simulation. The growth curves used for each type of neurite in these simulations are derived in the next sections. The network is initially balanced to its low frequency AI firing state ($t < 1500$ s in Figs 3B, 3C, 4A and 4B and panel 1 in Figs 3A and 4C). The network in this state represents a physiologically functioning cortical network, and has also been demonstrated by Vogels, Sprekeler et al. to function as a store for attractor-less associative memories [39]. Structural plasticity is then enabled at all neurites, and it is confirmed that the network maintains its balanced state under the combined action of the two homeostatic mechanisms ($1500$ s $< t < 2000$ s in Figs 3B, 3C, 4A and 4B). At ($t = 2000$ s), the network is deafferented by removing external inputs to neurons in the LPZ.

In line with experimental findings, the immediate result of deafferentation of the inhibition-balanced network is the loss of activity in neurons of the LPZ. For neurons outside the LPZ, on the other hand, our simulations show an increase in activity suggesting a net loss of inhibition rather than excitation ($t = 2000$ s in Fig 3C) (~8% and ~19% increase in mean firing rate of E and I neurons in the peri-LPZ in the first 100 seconds after deafferentation respectively). To our knowledge, this phenomenon has not yet been investigated in experiments, and an increase in neuronal activity following deafferentation of a neighbouring area is therefore the first testable prediction provided by our model.

The change in activity caused by deafferentation stimulates neurite turnover in neurons of the network in accordance with our proposed activity dependent growth rules ($t > 2000$ s). Over time, activity is gradually restored in the network to pre-deafferentation levels ($t = 18,000$ s in Fig 3B and 3C, and panel 4 in Figs 3A and 4C).

Even though the mean activity of neurons within and outside the LPZ returns to pre-deprivation levels, in our simulations, network reorganization by structural plasticity leads to synchronous spiking in neurons in the LPZ instead of its normal AI firing ($t > 4000$ s in Fig 4A and 4B, and panels 3 and 4 in Fig 4C). This predicted effect of network rewiring on the temporal characteristics of neural activity should be an interesting subject for future experimental studies. Furthermore, the observed lack of AI activity in the LPZ is expected to have functional implications; this is another promising topic for future theoretical work.

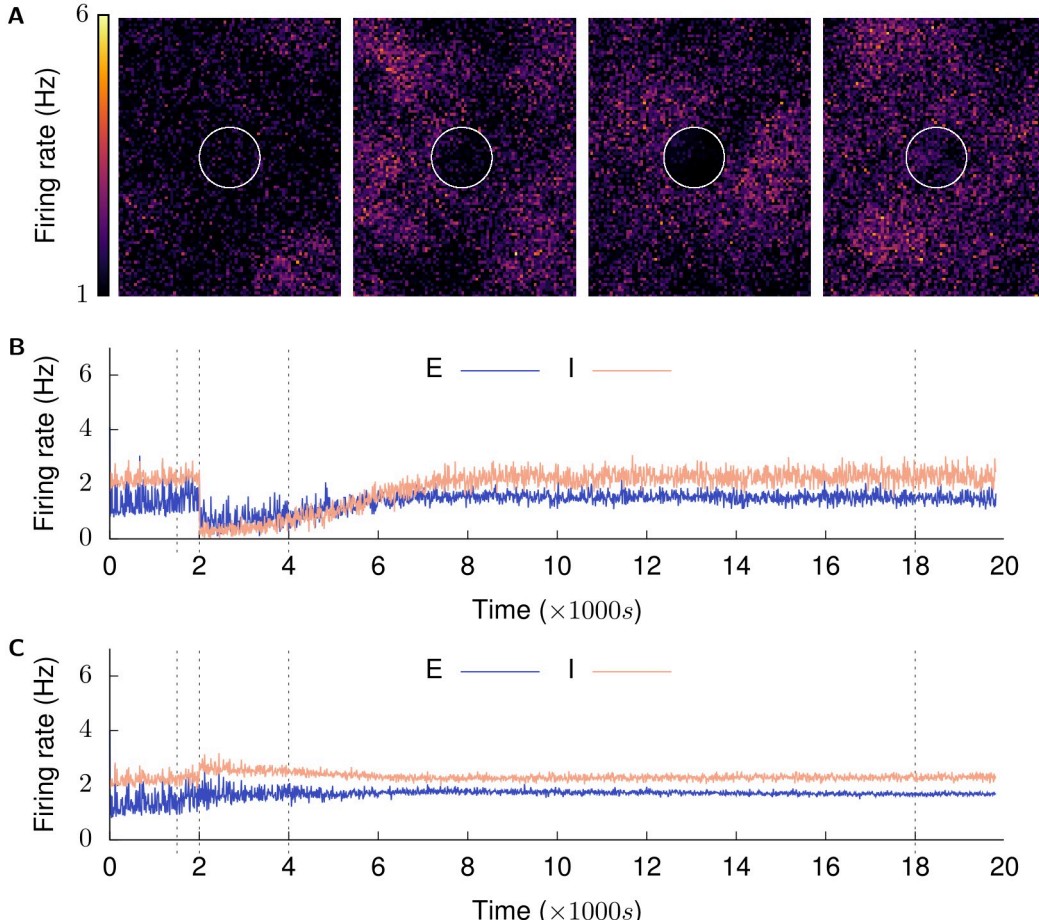

**Fig 3. Recovery of activity over time.** (Mean firing rates of neurons are calculated over a 2500 ms window and plotted in 100 ms increments.). **(A)** shows the firing rates of the whole excitatory population at $t = \{1500$ s, 2001.5 s, 4000 s, and 18,000 s$\}$. These are marked by dashed lines in the next graphs. The LPZ is indicated by white circles here and in following figures. **(B)** shows mean firing rate of neurons in LPZ-C; **(C)** shows mean firing rate of neurons in peri-LPZ; The network is permitted to achieve its balanced Asynchronous Irregular (AI) low frequency firing regime under the action of inhibitory synaptic plasticity ($t \leq 1500$ s). Structural plasticity is then activated at all neurites—pre-synaptic and post-synaptic, excitatory and inhibitory—to confirm that the network remains in its balanced AI state (panel 1 in A). At ($t = 2000$ s), neurons in the LPZ are deafferented (panel 2 in A is at $t = 2001.5$ s) and the network allowed to repair itself under the action of our structural plasticity mechanism (panels 3 ($t = 4000$ s) and 4 ($t = 18,000$ s) in A).

## Activity-dependent dynamics of post-synaptic structures

The previous section provides an overview of the network model where structural plasticity, governed by the different growth rules for different neurites, is active at all neurites. Growth rules for each type of neurite, however, were derived sequentially. Since the activity of neurons depends on the inputs received through their post-synaptic neurites, we first derived the growth rules for these neurites.

All neurons in the LPZ, excitatory and inhibitory, show near zero activity after deafferentation due to a net loss in excitatory input (panel 2 in Figs 3A and 4C, and $t = 2000$ s in Fig 3B). Experimental studies report that these neurons gain excitatory synapses on newly formed dendritic spines [28] and lose inhibitory shaft synapses [11] to restore activity after deprivation. The increase in lateral excitatory projections to these neurons requires them to gain excitatory dendritic (post-synaptic) elements to serve as contact points for excitatory axonal collaterals.

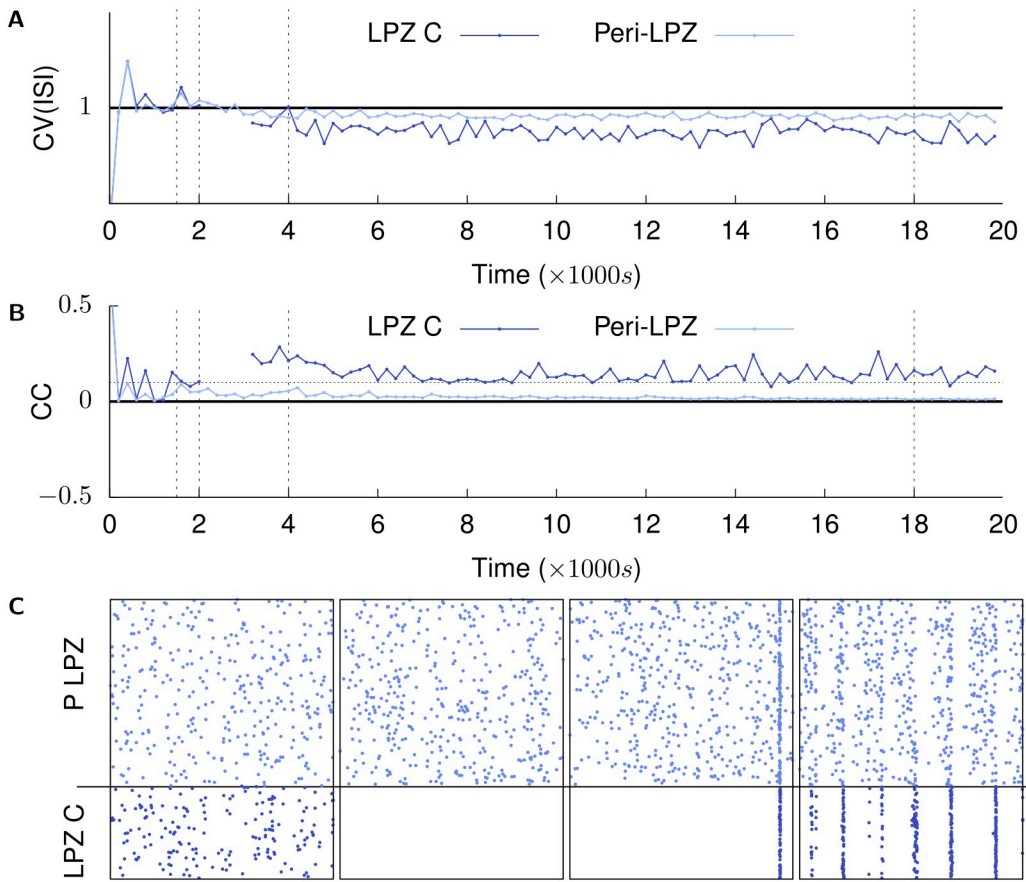

**Fig 4. Recovery of activity over time: population firing characteristics. (A)** shows the coefficient of variation (*CV*) of the inter-spike intervals of neurons in the LPZ-C and peri-LPZ. **(B)** shows the average pairwise cross-correlation between neurons in the LPZ-C and peri-LPZ calculated over 5 ms bins [43]. (dotted horizontal line is *y* = 0.1); **(C)** shows spike times of neurons in the LPZ C and peri-LPZ over a 1 s period at *t* = {1500 s, 2001.5 s, 4000 s, 18,000 s}. The network is permitted to achieve its balanced Asynchronous Irregular (AI) low frequency firing regime under the action of inhibitory synaptic plasticity (*t* ≤ 1500 s). At (*t* = 2000 s), neurons in the LPZ are deafferented (panel 2 in B is at *t* = 2001.5 s) and the network allowed to repair itself under the action of our structural plasticity mechanism (panels 3 (*t* = 4000 s) and 4 (*t* = 18,000 s) in B). As can be seen here, the network does not return to its AI state after repair (graphs are discontinuous because ISI CV and CC are undefined in the absence of spikes).

At the same time, inhibitory synapses can be lost by the retraction of inhibitory dendritic elements. This suggests that new excitatory post-synaptic elements should be formed and inhibitory ones removed when neuronal activity is less than its optimal level (($[Ca^{2+}] < \psi$ in Fig 5A):

$$\frac{dz_{post}^{E}}{dt} > 0 \qquad \text{for} \quad [Ca^{2+}] < \psi$$

$$\frac{dz_{post}^{I}}{dt} < 0 \qquad \text{for} \quad [Ca^{2+}] < \psi$$

(4)

While we were unable to find experimental evidence on the activity of excitatory or inhibitory neurons just outside the LPZ, in our simulations, these neurons exhibit increased activity after deafferentation (*t* = 2000 s in Fig 3C). Unlike neurons in the LPZ that suffer a net loss of excitation, these neurons appear to suffer a net loss of inhibition, which indicates that they

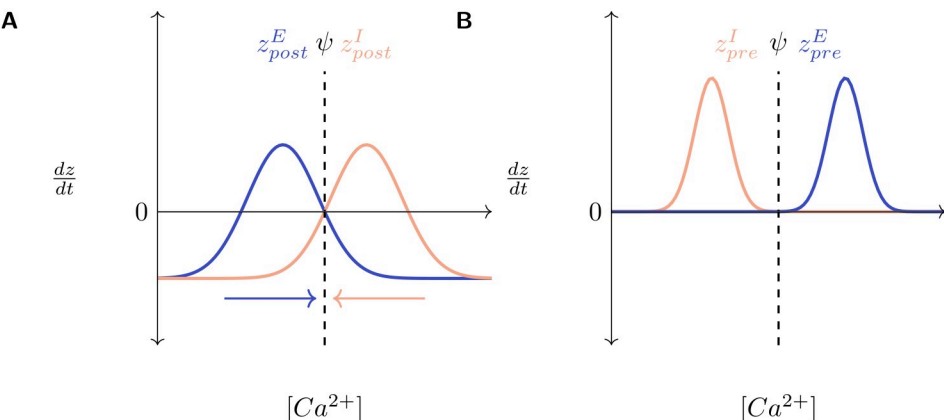

**Fig 5. Activity-dependent dynamics of synaptic elements (*dz/dt*) as functions of a neuron's time averaged activity ([*Ca²⁺*]). (A) post-synaptic elements**: Post-synaptic elements of a neuron react to deviations in activity from the optimal level ($\psi$) by countering changes in its excitatory or inhibitory inputs to restore its E-I balance. For both excitatory and inhibitory neurons, excitatory post-synaptic elements sprout when the neuron experiences a reduction in its activity, and retract when the neuron has received extra activity. Thus, the stable fixed point of the growth curve for post-synaptic excitatory neurites is found at $\epsilon_{post}^E$ where the slope of the growth curve is negative. Inhibitory post-synaptic elements for all neurons follow the opposite rule: they sprout when the neuron has extra activity and retract when the neuron is deprived of activity. The stable fixed point of the growth curve for post-synaptic inhibitory neurites is therefore, found at $\eta_{post}^I$ where the slope of the growth curve is positive. Together, these growth curves ensure that when the neuron has more than optimal activity, it will lose excitation and attempt to gain inhibition to reduce its net activity (red arrow). Similarly, the neuron attempts to gain excitation and loses inhibition to gain net activity when it has less than optimal activity (blue arrow). The optimal activity level, $\psi$, thus acts as the stable fixed point for both post-synaptic growth curves. **(B) pre-synaptic elements**: The connectivity of the network also depends on the pre-synaptic connectivity. Specifically, neurons attempting to gain synapses can only do so if free neurites of the required type are available. In excitatory neurons, axonal sprouting is stimulated by extra activity. In inhibitory neurons, on the other hand, deprivation in activity stimulates axonal sprouting. Synaptic elements that do not find corresponding partners to form synapses (free synaptic elements) decay exponentially with time. These graphs are illustrations of the regimes that the growth curves must follow.

must gain inhibitory and lose excitatory inputs to return to their balanced state. Hence, the formation of new inhibitory dendritic elements and the removal of their excitatory counterparts occurs in a regime where neuronal activity exceeds the required amount (($[Ca^{2+}] > \psi$) in Fig 5A):

$$\frac{dz_{post}^E}{dt} < 0 \qquad \text{for} \quad [Ca^{2+}] > \psi$$

$$\frac{dz_{post}^I}{dt} > 0 \qquad \text{for} \quad [Ca^{2+}] > \psi$$

(5)

The constraints described by equations Eqs (2), (4) and (5) can be satisfied by families of Gaussian growth rules for excitatory and inhibitory dendritic elements, with $\epsilon_{post}^E = \psi$ and $\eta_{post}^I = \psi$, respectively (Fig 5A and Table 1). $\epsilon_{post}^E$ and $\eta_{post}^I$ individually represent the stable fixed points for the excitatory and inhibitory post-synaptic growth curves respectively, and their intersection at the neuron's optimal activity level, $\psi$, ensures that deviations to the neuron's activity away from $\psi$ are countered by the turnover of the neuron's post-synaptic elements. This is further illustrated in the next section. Note that whereas the slope of the growth curve for excitatory post-synaptic neurites is negative at its stable fixed point $\epsilon_{post}^E$, the slope of the growth curve for inhibitory post-synaptic elements at $\eta_{post}^I$ is positive because the activity of the neuron is inversely related to the number of inhibitory post-synaptic inputs it receives.

**Table 1. Growth curves for synaptic elements.**

| | Post-synaptic elements | | | | | |
| --- | --- | --- | --- | --- | --- | --- |
| | Excitatory | | | Inhibitory | | |
| | $(\epsilon = \psi)$ | $(\eta < \psi < \epsilon)$ | $(\psi = \eta)$ | $(\epsilon = \psi)$ | $(\eta < \psi < \epsilon)$ | $(\psi = \eta)$ |
| Normal Repair | Yes | No | Yes | Yes | No | Yes |
| | Yes | NA | No | No | NA | Yes |
| | Pre-synaptic elements | | | | | |
| | Excitatory | | | Inhibitory | | |
| | $(\epsilon = \psi)$ | $(\eta < \psi < \epsilon)$ | $(\psi = \eta)$ | $(\epsilon = \psi)$ | $(\eta < \psi < \epsilon)$ | $(\psi = \eta)$ |
| Normal Repair | Yes | No | Yes | Yes | No | Yes |
| | No | NA | Yes | Yes | NA | No |

Only the derived families of pre- and post-synaptic growth curves allowed for both: (a) stable function of network in the absence of any deafferentation; (b) restoration of activity to the LPZ by an inward propagation of excitatory connections and an outward growth of inhibitory projections. (NA = Not applicable: since these regimes did not maintain a normal (without deafferentation) network in its balanced state, they were not tested for successful repair.)

Figs 6 and 7 show the course of rewiring of excitatory and inhibitory connections to excitatory neurons in the centre of the LPZ from simulations using growth curves satisfying the derived constraints. As described in experimental studies, the loss of activity by neurons in the LPZ is followed by an increase in excitatory input connections [13, 28] and a transient reduction in inhibitory input connections [11]. Specifically, as also found in these experiments, the increase in excitatory inputs is dominated by an ingrowth of lateral projections from outside the LPZ. Both of these features can be seen in Fig 6A and 6B. As shown in Figs 8 and 9, neurons directly outside the LPZ lose excitatory and gain inhibitory input connections to reduce their activity back to their optimal values. These figures also show that, in line with experimental observations, there is an increase in the number of inhibitory synapses these neurons receive from the LPZ. Finally, even though the much larger number of inhibitory neurons outside the LPZ do increase their inhibitory projections on to the network, Figs 7C and 9C confirm that the strongest inhibitory projections are received from the LPZ.

**Post-synaptic growth rules stabilise individual neurons.** Experimental evidence suggests that not just networks, but also individual neurons in the brain maintain a finely tuned balance between excitation and inhibition (E-I balance) [44–46]. Even though the configurations for post-synaptic growth rules were derived here from network level observations, we wondered if the complementary nature of the excitatory and inhibitory post-synaptic growth rules could serve a homeostatic purpose at the level of individual neurons by stabilising their activity.

To test this, we modelled a neuron in isolation to investigate how its input connectivity would be affected by changes in activity as per our post-synaptic growth curves (Fig 10A). The neuron is initialised with an input connectivity similar to a neuron from the network in its steady state: it has the same number of excitatory ($z_{post}^E$) and inhibitory ($z_{post}^I$) dendritic elements and receives the same mean conductances through them ($g_{EE}, g_{IE}$). Thus, the $[Ca^{2+}]$ of the neuron in this state represents its optimal activity ($\psi = [Ca^{2+}]$ at $t = 0$ s in Fig 10B). In this scenario, the net input conductance received by the neuron ($g_{net}$), which modulates its activity, can be estimated as the difference of the total excitatory ($g_E$) and inhibitory ($g_I$) input conductances.

The activity of the neuron is then varied by an external sinusoidal current stimulus (Fig 10B). In addition, the deviation of the neuron's excitatory ($\Delta g_E$), inhibitory ($\Delta g_I$), and net input conductance ($\Delta g_{net}$) from baseline levels due to the formation or removal of dendritic elements under the action of the growth curves is recorded (Fig 10C). We find that that modifications of

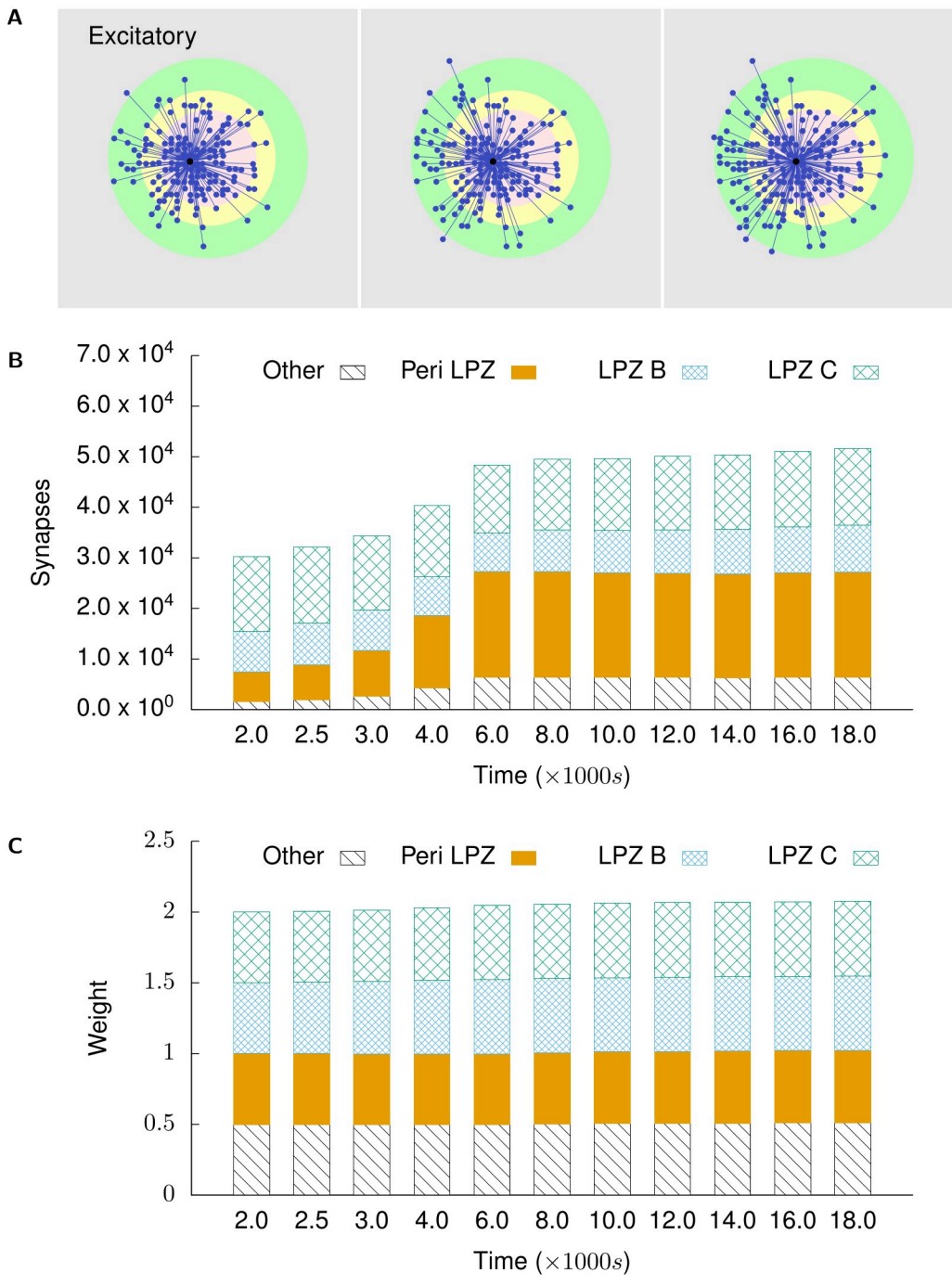

**Fig 6. Excitatory projections to excitatory neurons in the centre of the LPZ. (A)** shows incoming excitatory projections to a randomly chosen neuron in the centre of the LPZ at different stages of our simulations. From left to right: $t = 2000$ s, $t = 4000$ s, and $t = 18,000$ s. **(B)** shows the total numbers of incoming excitatory projections to neurons in the centre of the LPZ from different regions at different points in time. **(C)** shows the mean weight of projections received by neurons in the centre of the LPZ from different regions at different points in time. Following our proposed growth rules for post-synaptic elements and consistent with experimental reports, the deprived neurons in the LPZ C gain lateral excitatory inputs from neurons outside the LPZ.

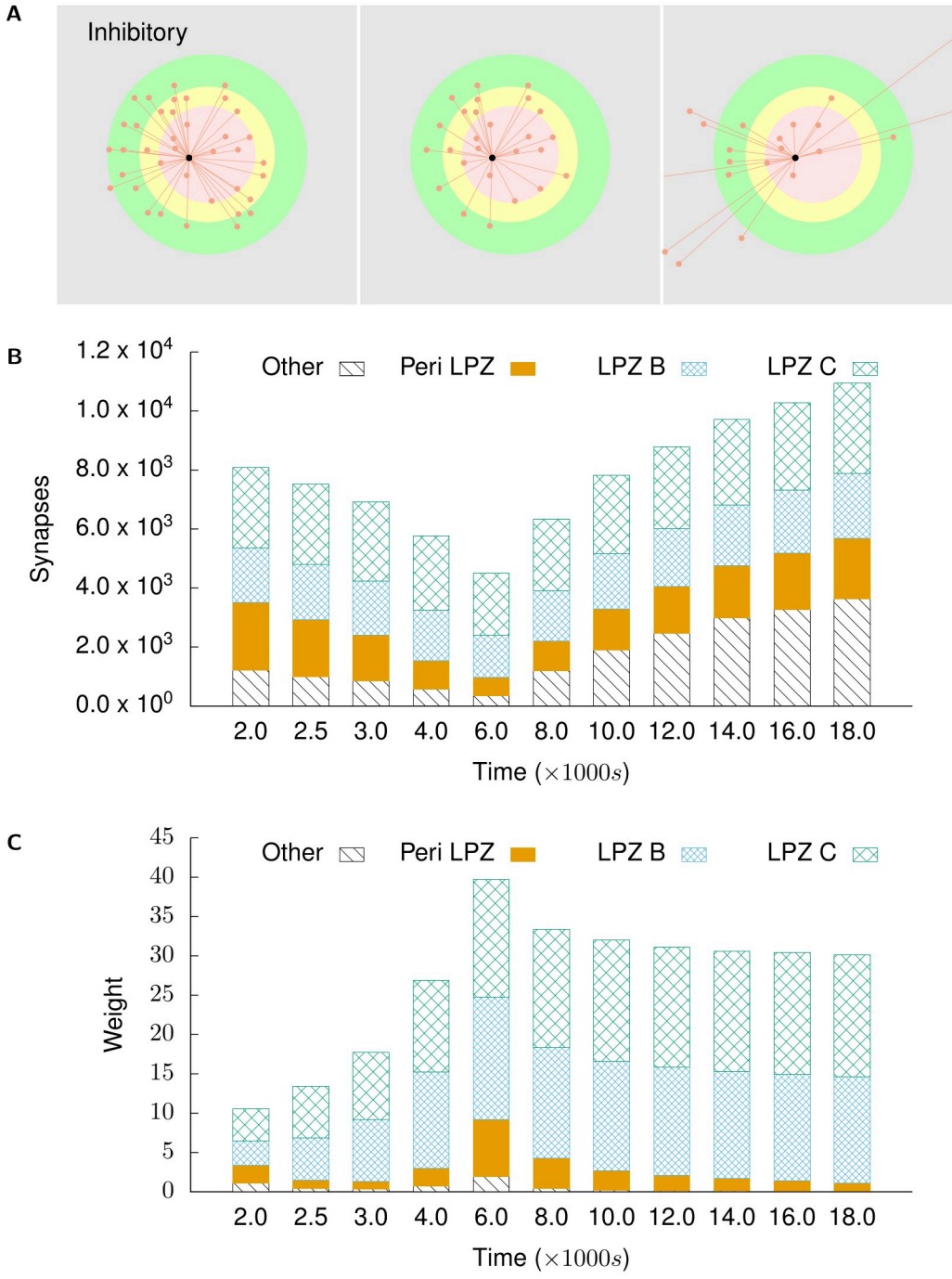

**Fig 7. Inhibitory projections to excitatory neurons in the centre of the LPZ. (A)** shows incoming inhibitory projections to a randomly chosen neuron in the centre of the LPZ at different stages of our simulations. From left to right: $t$ = 2000 s, $t$ = 4000 s, and $t$ = 18,000 s. **(B)** shows the total numbers of incoming inhibitory projections to neurons in the centre of the LPZ from different regions at different points in time. **(C)** shows the mean weight of projections received by neurons in the centre of the LPZ from different regions at different points in time. Also in line with biological observations, they temporarily experience dis-inhibition after deafferentation. However, as these neurons gain activity from their new lateral excitatory inputs, the number of their inhibitory input connections increases again in order to restore the E-I balance.

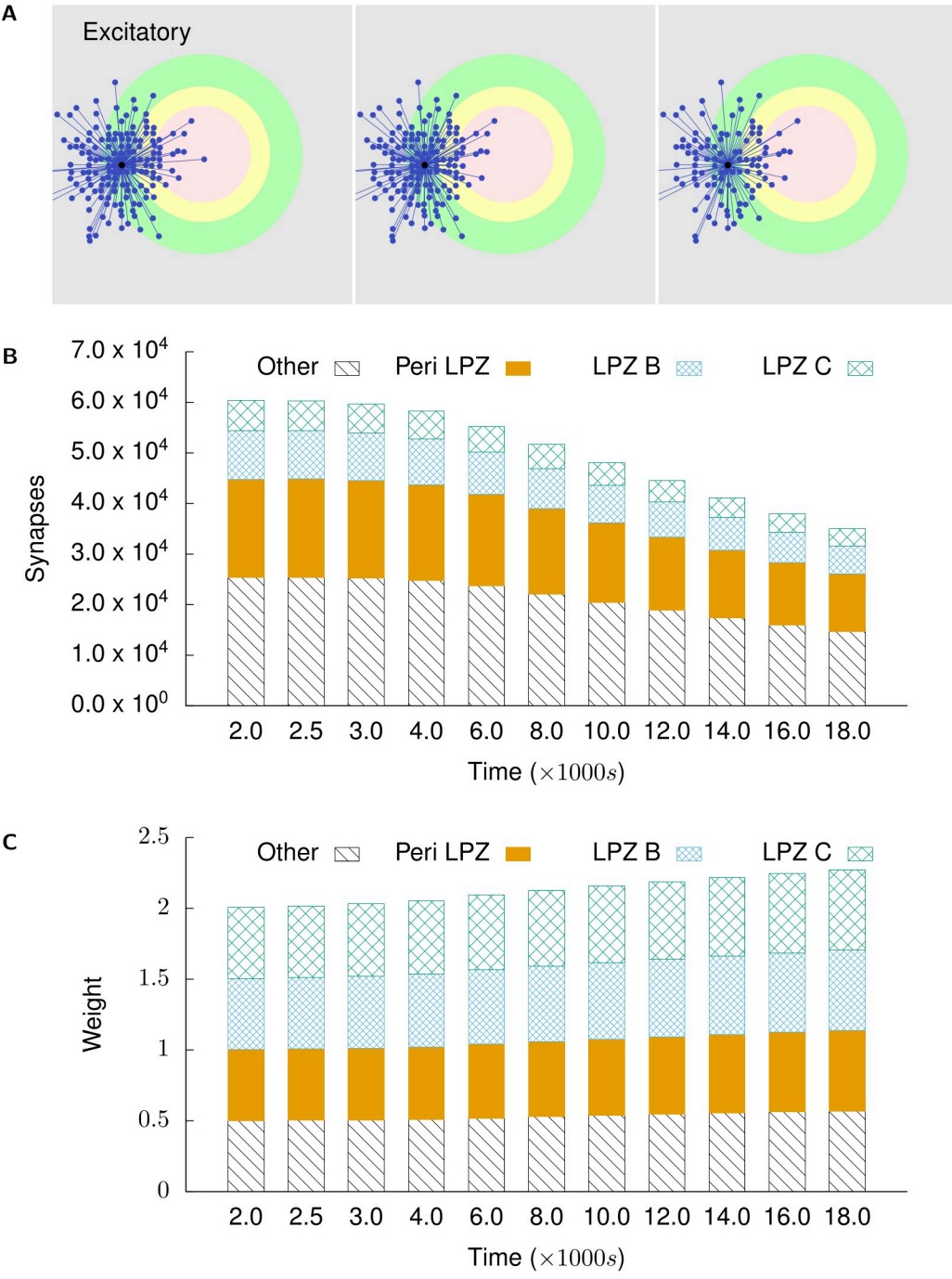

**Fig 8. Excitatory projections to excitatory neurons in the peri-LPZ.** (A) shows incoming excitatory projections to a randomly chosen neuron in the peri-LPZ at different stages of our simulations. From left to right: $t = 2000$ s, $t = 4000$ s, and $t = 18,000$ s. (B) shows the total numbers of incoming excitatory projections to neurons in the peri-LPZ from different regions at different points in time. (C) shows the mean weight of projections received by neurons in the peri-LPZ from different regions at different points in time. In contrast to neurons in the LPZ, neurons outside the LPZ experience an increase in activity in our simulations. As a result of our growth rules, these neurons lose excitatory inputs.

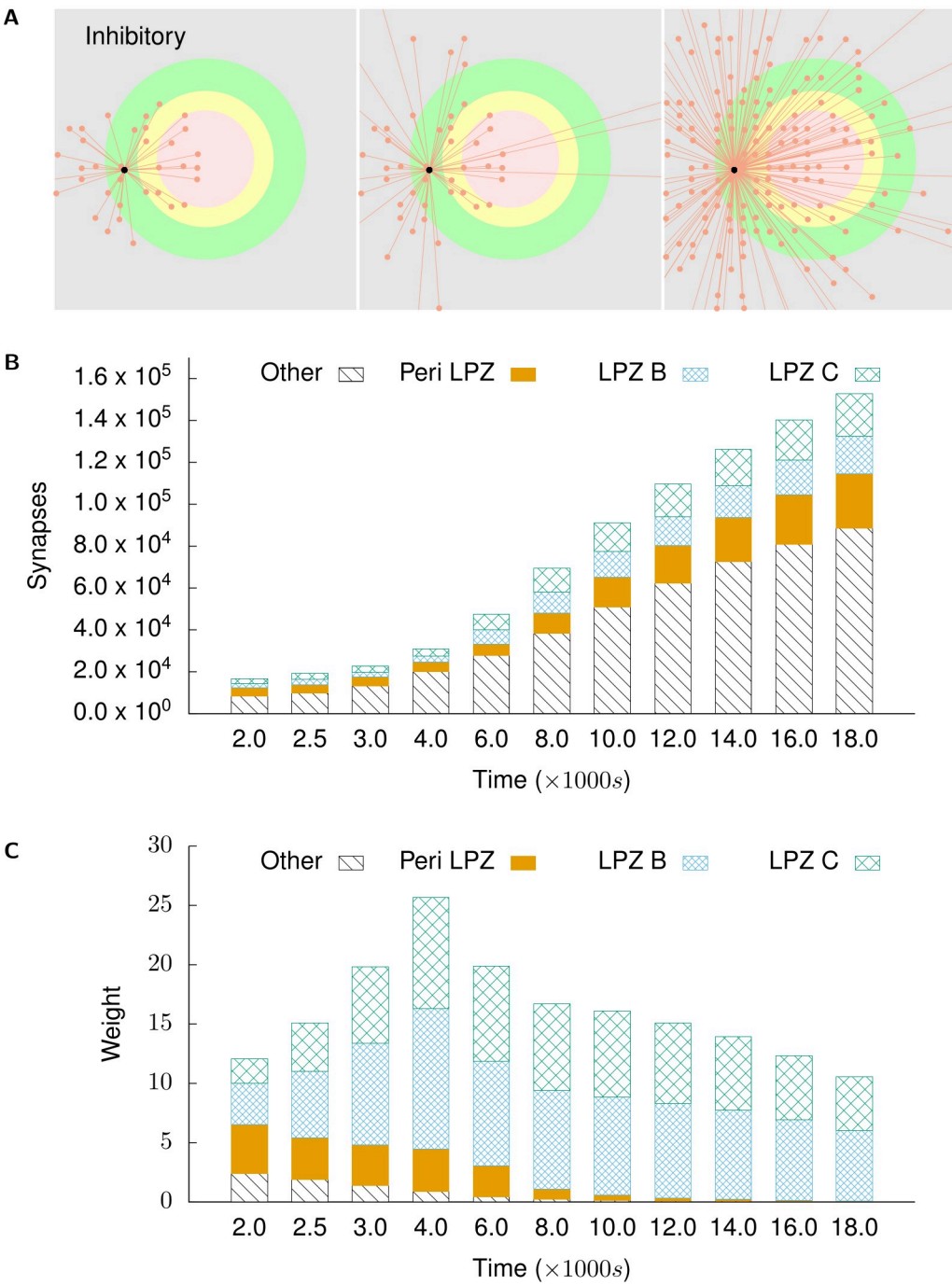

**Fig 9. Inhibitory projections to excitatory neurons in the peri-LPZ. (A)** shows incoming inhibitory projections to a randomly chosen neuron in the peri-LPZ at different stages of our simulations. From left to right: $t$ = 2000 s, $t$ = 4000 s, and $t$ = 18,000 s. **(B)** shows the total numbers of incoming excitatory projections to neurons in the peri-LPZ from different regions at different points in time. **(C)** shows the mean weight of projections received by neurons in the peri-LPZ from different regions at different points in time. In contrast to neurons in the LPZ, neurons outside the LPZ experience an increase in activity in our simulations. As a result of our growth rules, these neurons gain inhibitory inputs.

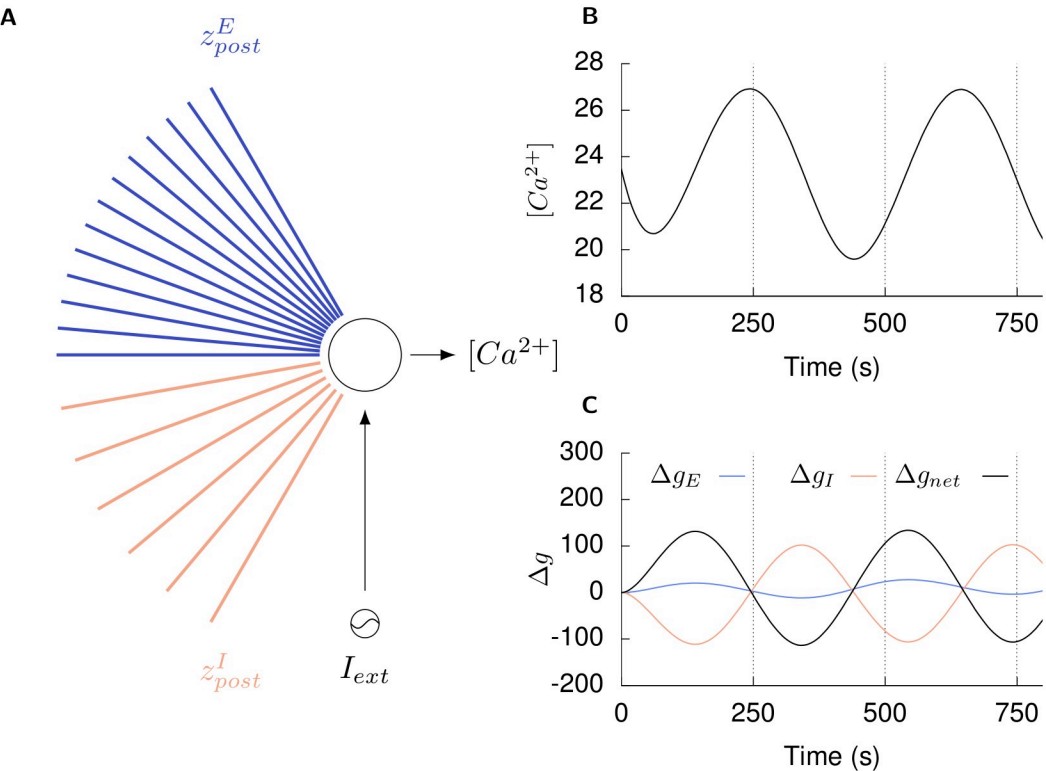

**Fig 10. Single neuron simulations show the homeostatic effect of the post-synaptic growth rules. (A)** A neuron in its balanced state receives excitatory ($g_E$) and inhibitory ($g_I$) conductance inputs through its excitatory ($z_{post}^E$) and inhibitory ($z_{post}^I$) dendritic elements, respectively, such that its activity ($[Ca^{2+}]$) is maintained at its optimal level ($\psi$) by its net input conductance ($g_{net}$). **(B)** An external sinusoidal current stimulus ($I_{ext}$) is applied to the neuron to vary its activity from the optimal level. **(C)** Under the action of our post-synaptic growth curves, the neuron modifies its dendritic elements to change its excitatory ($\Delta g_E$) and inhibitory ($\Delta g_I$) conductance inputs such that the net change in its input conductance ($\Delta g_{net}$) counteracts the change in its activity: an increase in $[Ca^{2+}]$ due to the external stimulus is followed by a decrease in net input conductance received through the post-synaptic elements and vice versa (dashed lines in B and C).

the input connectivity of the neuron result in alterations to its excitatory and inhibitory input such that the net change in its input conductance counteracts alterations in its activity: an increase in $[Ca^{2+}]$ due to the external stimulus is followed by a decrease in net input conductance through the post-synaptic elements and vice versa (dashed lines in Fig 10B and 10C).

These simulation results show that even though the activity dependent growth rules of excitatory and inhibitory post-synaptic elements are derived from network simulations, they also serve to stabilise activity in single neurons by helping to maintain the balance between their excitatory and inhibitory inputs. Here, we note that since structural plasticity is modelled as the discrete formation and removal of whole synapses with pre-set conductances in the MSP framework that is used here, it causes relatively large perturbations in the excitatory and inhibitory levels of both individual neurons and the network. As we discuss in later sections, synaptic plasticity plays a critical role in fine tuning conductances to enable the network to achieve E-I balance.

## Activity dependent dynamics of pre-synaptic structures

While the activity dependent formation and degradation of post-synaptic elements provides a homeostatic mechanism for the stabilisation of activity in single neurons and the network, the

increase in excitatory or inhibitory input received by a neuron also relies on the availability of pre-synaptic counterparts. Next, we derive activity dependent growth rules for excitatory ($z^E_{pre}$) and inhibitory ($z^I_{pre}$) pre-synaptic elements in a similar manner to that used for post-synaptic elements.

Within the LPZ, the increase in excitation requires a corresponding increase in the supply of excitatory pre-synaptic elements. Experimental evidence reports a sizeable increase in the formation and removal of axonal structures in and around the LPZ [27], with a marked addition of lateral projections from neurons outside the LPZ into it [6]. While an increase in pre-synaptic elements within the LPZ may contribute to repair, an inflow of activity from the periphery of the LPZ to its centre has been observed in experiments [6, 20, 28], pointing to the inwards sprouting of excitatory axonal projections from outside the LPZ as the major driver of homeostatic rewiring. For this sprouting of excitatory projections from the non-deafferented area into the LPZ to take place in our simulations, the increase in activity in neurons outside the LPZ must stimulate the formation of their excitatory axonal elements:

$$\frac{dz^E_{pre}}{dt} > 0 \qquad \text{for} \quad [Ca^{2+}] > \psi \tag{6}$$

Conversely, neurons outside the LPZ with increased activity need access to inhibitory pre-synaptic elements in order to receive the required additional inhibitory input. Deafferentation studies in mouse somatosensory cortex [6] report more than a 2.5 fold increase in the lengths of inhibitory axons projecting out from inhibitory neurons in the LPZ two days after the peripheral lesion. This outgrowth of inhibitory projections preceded and was faster than the ingrowth of their excitatory analogues [6, 9]. In our simulations, the experimentally observed outward protrusion of inhibitory axons from the LPZ requires that the formation of inhibitory pre-synaptic elements is driven by reduced neuronal activity:

$$\frac{dz^I_{pre}}{dt} > 0 \qquad \text{for} \quad [Ca^{2+}] < \psi \tag{7}$$

To further validate the derived pre-synaptic growth curves, shown in Fig 5B and Table 1, the complete set of possible configurations of pre-synaptic growth curves was tested. These are labelled **G0, G1, G2, G3, G4**, and **G5** and illustrated in Fig 11:

- **G0**: control case where there are no growth curves, achieved by setting $v = 0$,

- **G0'**: constant axonal sprouting, irrespective of neuronal activity $v > 0$,

- **G1**: both inhibitory and excitatory axons sprout when activity is more than required,

- **G2**: (the selected growth curves shown in Fig 5B fall into this category) inhibitory axons sprout when activity is less than optimal, but excitatory axons sprout when activity is more than required,

- **G3**: excitatory axons sprout when activity is less than optimal, but inhibitory axons sprout when activity is more than required,

- **G4**: both excitatory and inhibitory axons sprout at optimal activity, and

- **G5**: both inhibitory and excitatory axons sprout when activity is less than optimal.

As summarised in Table 2, only the derived configuration for pre-synaptic growth curves reproduced all experimentally reported features of the repair process: inhibitory axons sprout when neuronal activity is less than optimal, but excitatory axons sprout when activity is more than required. Since the activity of neurons is continuously stabilised by the previously derived

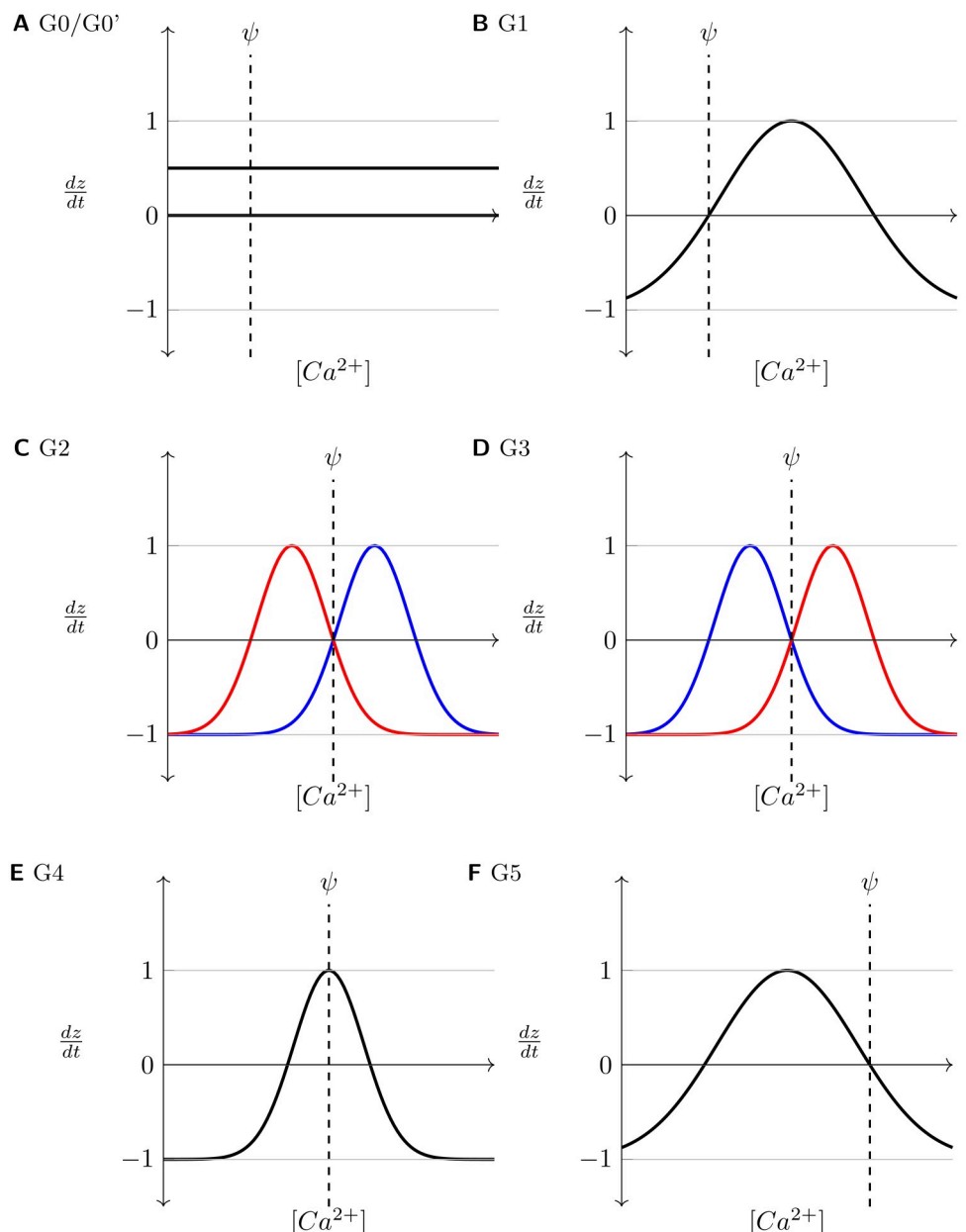

**Fig 11. Axonal growth curves investigated in the study (where applicable, Red: Inhibitory, Blue: Excitatory).** Only growth curves shown in C/G2 reproduce the course of repair observed in experiments (Table 2). **G0**: no growth curves (no sprouting or retraction); **G0'**: constant axonal sprouting, irrespective of neuronal activity; **G1**: both inhibitory and excitatory axons sprout when activity is more than required; **G2**: inhibitory axons sprout when activity is less than optimal, but excitatory axons sprout when activity is more than required; **G3**: excitatory axons sprout when activity is less than optimal, but inhibitory axons sprout when activity is more than required; **G4**: both excitatory and inhibitory axons sprout at optimal activity. **G5**: both inhibitory and excitatory axons sprout when activity is less than optimal.

post-synaptic growth curves, the turnover of pre-synaptic elements is also kept in check. Thus, as neurons in the network achieve their optimal activity levels, the turnover of both post- and pre-synaptic elements ceases.

Similar to the post-synaptic growth rules, the pre-synaptic growth rules for excitatory and inhibitory neurons were also treated separately and their parameters were tuned iteratively

**Table 2. Summary of axonal growth curves tested in the model.**

| | G0 | G0' | G1 | G2 | G3 | G4 | G5 |
|---|---|---|---|---|---|---|---|
| Network initially remains stable | Y | Y | Y | Y | Y | N | Y |
| Neurons in LPZ gain required activity | Y | Y | Y | Y | N | NA | N |
| Neurons outside LPZ lose extra activity | Y | Y | Y | Y | NA | NA | NA |
| Network returns to balanced state | N | Y | N | Y | NA | NA | NA |
| LPZ B restores before LPZ C | NA | N | NA | Y | NA | NA | NA |
| Ingrowth of E projections into LPZ | NA | N | NA | Y | NA | NA | NA |
| Outgrowth of I projections from LPZ | NA | N | NA | Y | NA | NA | NA |
| Dis-inhibition in LPZ | NA | NA | NA | Y | NA | NA | NA |

Each row represents a feature that is observed in experiments: **1. Network initially remains stable**: the network should remain stable without deafferentation; **2. Neurons in LPZ gain activity**: increase in activity of LPZ neurons to pre-deafferentation levels; **3. Neurons outside LPZ lose activity**: decrease in activity of neurons outside the LPZ to pre-deafferentation levels; **4. Network returns to balanced state**: the network should return to its balanced stable state after activity of all neurons has been restored to pre-deafferentation levels; **5. LPZ B restores before LPZ C**: activity should be restored to the LPZ B neurons before the LPZ C neurons; **6. Ingrowth of axonal projections into LPZ**: there should be ingrowth of excitatory axons to the LPZ; **7. Outgrowth of inhibitory projections from LPZ**: outgrowth of inhibitory axons from the LPZ should stabilise neurons outside the LPZ; **8. Dis-inhibition in LPZ**: dis-inhibition should be observed due the LPZ neurons. Each column represents a set of growth curves (illustrated in Fig 11).

over repeated simulations. Since inhibitory neurons form only one-fourth of the neuronal population, and only a small number of these fall into the LPZ, in this study, simulations require the growth rates of inhibitory axonal elements to be high enough to re-balance the large number of neurons outside the LPZ that have higher than optimal activity. If the growth rate of inhibitory pre-synaptic elements is not high enough, newly sprouted inhibitory post-synaptic elements on these neurons will remain unconnected. Without the additional inhibition, these neurons will rely solely on the loss of excitatory synapses by the retraction of their excitatory post-synaptic elements to reduce their activity back to optimal levels. A reduction in the excitatory connectivity of the network may adversely affect network functions by disrupting Hebbian assemblies stored in them.

Fig 12A and 12B show the rewiring of axonal projections from an excitatory neuron in the peri-LPZ and an inhibitory neuron in the centre of the LPZ, respectively. Following the growth rules derived above, our simulations correctly reproduce the inward sprouting of excitatory axons into the LPZ and the outward sprouting of inhibitory axons from the LPZ that is observed during the repair process.

## Synaptic and structural plasticity both contribute to network repair

In all our previous simulations, the network rewiring after deafferentation of the LPZ occurred in the presence of both activity-dependent structural plasticity and inhibitory synaptic plasticity. In order to study the functional role of the two plasticity mechanisms in the homeostatic regulation of activity after peripheral lesions, we simulated our model with each the mechanisms enabled in isolation (see Methods).

Results from our simulations where structural plasticity is disabled show that inhibitory synaptic plasticity alone, while able to re-balance neurons outside the LPZ by increasing the strength of their inhibitory inputs, fails to restore activity in the deprived neurons in the LPZ even after small peripheral lesions (Fig 13A and 13D). Although the homeostatic inhibitory synaptic plasticity on its own leads to a reduction in conductances of the inhibitory synapses projecting onto neurons in the LPZ, this is not sufficient to reactivate them. The stabilisation of activity in the neurons outside the LPZ, however, is successful due to the strengthening of

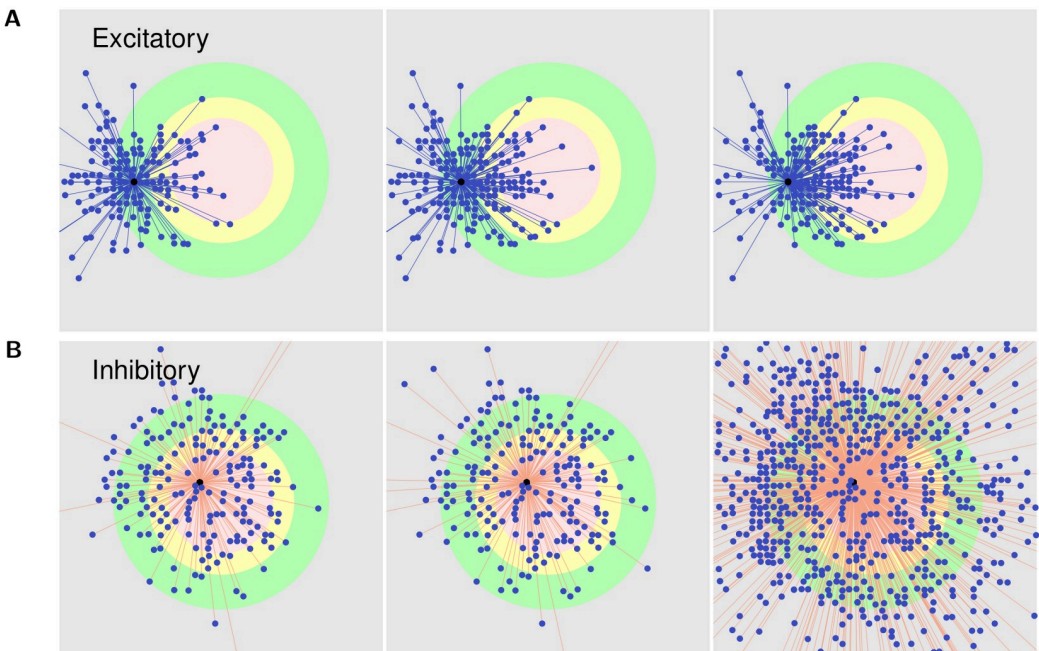

**Fig 12. Outgoing projections. (A)** shows the outgoing (axonal) projections of an excitatory neuron in the peri-LPZ. **(B)** shows the outgoing (axonal) projections of an inhibitory neuron in the LPZ C. From left to right: $t$ = 2000 s, $t$ = 4000 s, and $t$ = 18,000 s. As per our suggested growth rules for pre-synaptic elements, excitatory neurons produce new pre-synaptic elements and sprout axonal projections when they experience extra activity, while inhibitory neurons form new pre-synaptic elements and grow axons when they are deprived of activity. As a consequence and in line with experimental data, following deafferentation of the LPZ, excitatory neurons in the peri-LPZ sprout new outgoing projections that help transfer excitatory activity to neurons in the LPZ. Also in accordance with experimental work, inhibitory neurons inside the LPZ form new outgoing connections that transmit inhibition to neurons outside the LPZ.

IE synapses by STDP. In the absence of network rewiring by structural plasticity, this leads to a network where the neurons outside the LPZ retain their functionality while the LPZ is effectively lost. This indicates that the larger deviations from the desired activity that result from deafferentation in our balanced network model require the reconfiguration of network connectivity by structural plasticity to re-establish a functional balance.

Simulations where homeostatic synaptic plasticity was disabled, on the other hand, also failed to re-establish the balanced state of the network before the peripheral lesion (Fig 13C and 13D). While the activity of the deprived neurons in the LPZ initially increased back to pre-lesion levels, under the action of structural plasticity only, the network eventually started exhibiting abnormally high firing rates instead of settling in the desired low firing rate regime. This suggests that whereas structural plasticity does contribute to the restoration of E-I balance, the discrete and relatively large changes it makes to synaptic conductances are only able to bring the state of the network close to the desired balanced state in our model.

Thus, even though our numerical methods are insufficient to isolate the individual contributions of the two mechanisms, both homeostatic processes play a role in successful repair in our simulations—structural plasticity causes larger changes in network connectivity and synaptic plasticity fine tunes conductances to together establish stable activity in the network. These results are in line with the idea that multiple plasticity mechanisms may work in harmony to sustain functional brain networks at varying time scales.

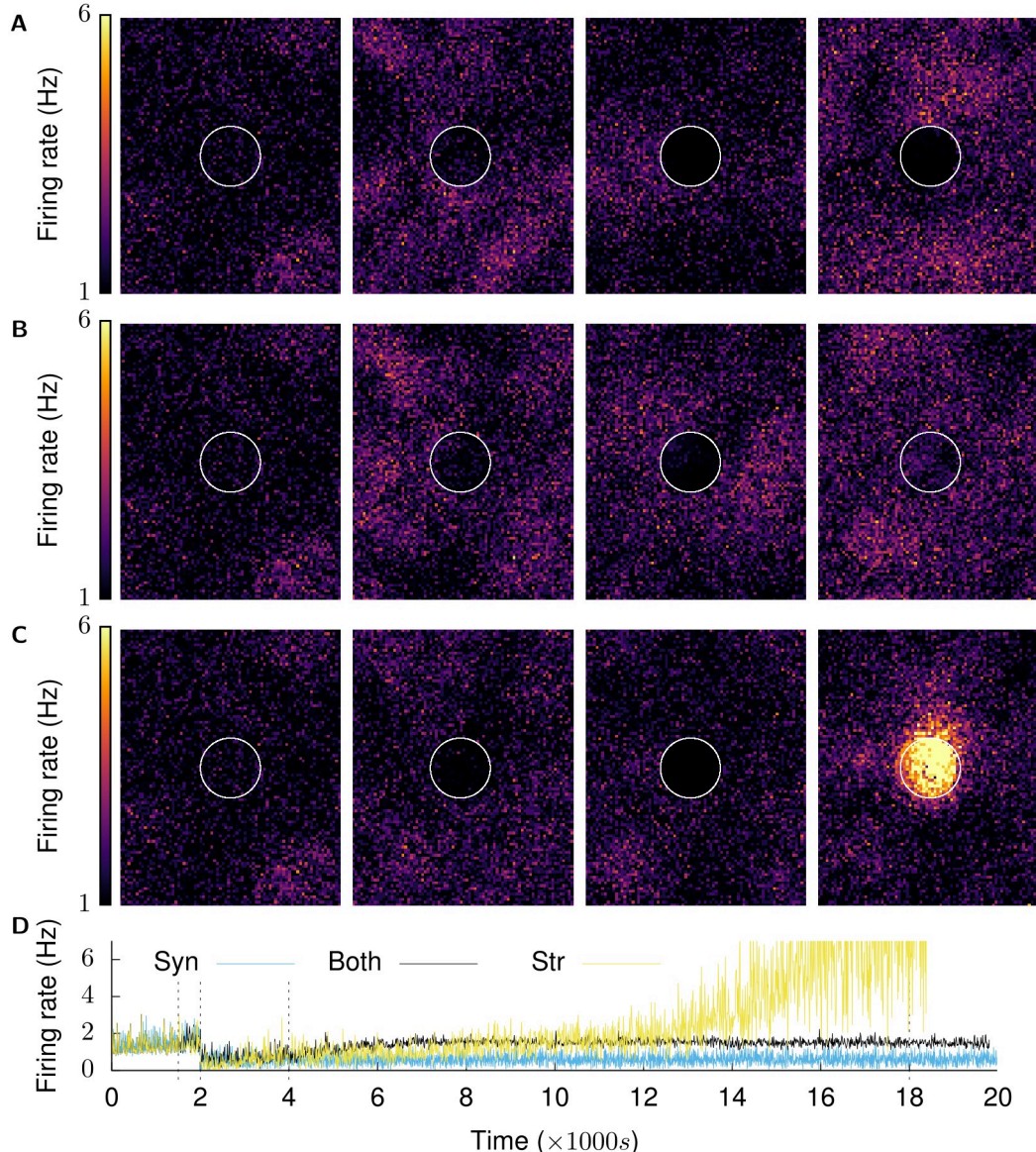

**Fig 13. Both structural and synaptic plasticity contribute to restoration of activity after deafferentation. (A), (B), (C)** show firing rate snapshots of neurons at $t$ = 1500 s, 2001.5 s, 4000 s, 18,000 s. **(A)** Synaptic plasticity only: after the network has settled in its physiological state by means of synaptic plasticity, structural plasticity is not enabled. With only synaptic plasticity present, the network is unable to restore activity to neurons in the LPZ. Neurons outside the LPZ return to their balanced state, but the neurons in the LPZ are effectively lost to the network. **(B)** Both structural and synaptic plasticity are enabled: neurons in the LPZ regain their low firing rate as before deafferentation. **(C)** Structural plasticity only: after the network has settled in its physiological state by means of synaptic plasticity, homeostatic synaptic plasticity is turned off and only structural plasticity is enabled. With only structural plasticity present, activity returns to neurons in the LPZ but does not stabilise in a low firing rate regime. The firing rate of these neurons continues to increase and, as a result, these neurons continue to turn over synaptic elements. This cascades into increased activity in neurons outside the LPZ, further causing undesired changes in network connectivity. **(D)** shows the mean population firing rates of neurons in the centre of the LPZ for the three simulation configurations. (Panel 1 is identical in all three simulation configurations because the same parameters are used to initialise all simulations.).

## Discussion

A better understanding of the factors that influence dynamic alterations in the morphology and connectivity of neuronal axons and dendrites is necessary to improve our knowledge of

the processes that shape the development and reorganisation of neuronal circuitry in the adult brain. Building on previous work [33], we present a new, spiking neural network model of peripheral lesioning in a biologically plausible cortical network model (Figs 1 and 2). We show that our simulations reproduce the course of changes in network connectivity as reported in experimental work (Fig 3), and we provide a number of testable predictions.

First, our model suggests that deafferentation does not necessarily result in the loss or even a decrease of activity in all neurons of the network. In our inhibition-balanced cortical network, neurons outside the LPZ experience a gain in activity because of a net loss in inhibition. This prediction should be tested in future experiments that investigate neuronal activity on the outer periphery of the LPZ.

Secondly, our model suggests that while the network may restore its mean activity, the temporal fine structure of the activity, and in particular its AI firing characteristic are permanently disturbed by deafferentation. This change in firing patterns of the network also merits experimental validation, especially given its implications for network function. Given that the inhibitory STDP mechanism is unable to maintain the network in its AI regime following repair by structural plasticity, the deviation from the AI firing regime is likely caused by the alteration of network connectivity during the repair process. Indeed, as Fig 6 shows, neurons in the central region of the LPZ gain a significant number of lateral excitatory connections from excitatory connections from neurons outside the LPZ ($3 \times 10^4$ before deafferentation at $t = 2000s$ vs $5 \times 10^4$ at the end of the repair process at $t = 18000s$) greatly increasing their excitatory input connectivity. This is in line with previous work that indicates that synchronisation may occur in networks of excitatory and inhibitory neurons when the number of inputs being received by neurons is more than a critical value [40, 47–51]. The precise relationship between network sparsity and population firing dynamics in a network balanced by the inhibitory STDP mechanism used here, however, does not appear to have been ascertained yet.

Thirdly, as the main objective of our work, we suggest different regimes for growth rules for each type of neurite (Fig 5). Whereas derived from network lesion experiments that were not aimed at studying the relation between activity and neurite turnover [6, 9, 10, 13, 27–30], experimental evidence seems to support our proposals. Our growth rule for excitatory dendritic elements is coherent with results from experimental studies in hippocampal slice cultures. In their study, Richards et al. note that reduced neuronal activity resulted in the extension of glutamate receptor-dependent processes from dendritic spines of CA1 pyramidal neurons [52]. In another study, Müller et al. report the loss of excitatory synapses in hippocampal slice cultures after the application of convulsants [53]. We were unable to locate experimental literature on the activity mediated dynamics of post-synaptic elements on inhibitory neurons.

On the pre-synaptic side, axonal turnover and guidance has been investigated in much detail, and is known to be a highly complex process incorporating multiple biochemical pathways [54, 55]. Our hypothesis regarding excitatory pre-synaptic structures is supported by a report by Perez et al. who find that CA1 pyramidal cells, which become hyper-excitable following hippocampal kainate lesions, sprout excitatory axons that may contribute to the epileptiform activity in the region [56]. For inhibitory pre-synaptic elements, where our model correctly reproduces the outgrowth of inhibitory axons from the LPZ as observed in experiments [6], we refer to Schuemann et al. who report that enhanced network activity reduced the number of persistent inhibitory boutons [57] over short periods of time (30 minutes) in organotypic hippocampal slice cultures. However, these experiments also found that prolonged blockade of activity (over seven days) did not affect inhibitory synapses, contrary to the reports from peripheral lesion studies [11, 30]. A prediction from our simulations is that the rates of formation of inhibitory pre-synaptic elements were required to be much greater than

that of other neurites to arrest extra excitation in the network. Here, this requirement is borne out of the small proportion of inhibitory neurons that stabilise the activity of the complete neuronal population. This may not be necessary in the brain, however, where activity is stabilised by a multitude of homeostatic mechanisms [58].

Indirect evidence on the temporal evolution of inhibitory projections to neurons in the LPZ further supports the inhibitory growth rules in our model (Fig 7B). While an initial disinhibition aids recovery in these deprived neurons, as activity is restored, a subsequent increase in inhibition in our simulations re-establishes the E-I balance in the deafferented region. This is in line with evidence that the pharmacological reduction of inhibition restores structural plasticity in the visual cortex [59], and to the best of our knowledge, has not been reproduced by previous models of structural plasticity. Our simulations, therefore, support the proposed role of inhibition as control mechanism for the critical window for structural plasticity [15, 60–64].

Our simulation results do not imply that these are the only configurations of activity dependent growth rules that can underlie the turnover of neurites. Given the variety of neurons and networks in the brain, many configurations (and a variety of growth curves in each configuration) may apply to neurons. The results suggested here are hypothesised using an inhibition-balanced AI cortical network model, and so must be limited to such networks. As an example of a different configuration of growth curves that replicated repair in a different network model, Butz and van Ooyen's simulations proposed that all neurites are sprouted when neurons have less than optimal activity, and that the condition necessary for repair by an ingrowth of excitatory connections is that dendritic elements should be formed before axonal ones [33]. Also similar to Butz and van Ooyen [33], the exact parameters governing the growth curves in the suggested configurations remain to be ascertained either experimentally or by more detailed modelling as discussed below.

Finally, our simulation results indicate that the suggested post-synaptic growth rules, while derived from network simulations, can contribute to the stability of activity in an isolated individual neuron (Fig 10). Since structural plasticity and synaptic plasticity are not independent processes in the brain, this is not a wholly surprising result. Structural plasticity of the volumes of spines and boutons underlies the modulation of synaptic efficacy by synaptic plasticity. Thus, given that synaptic plasticity mechanisms can stabilise the firing of individual neurons [65, 66], it follows that structural plasticity mechanisms could also be involved. Further, extending from the functional coupling of synaptic and structural plasticity, our simulations also require both structural and synaptic plasticity to be enabled for successful network repair (Fig 13). Thus, our simulation results lend further support to the notion that multiple plasticity mechanisms function in a cooperative manner in the brain at different temporal and spatial scales. The interaction between these two homeostatic mechanisms and its effect on stability of the individual neurons and network activity remains an important open question that cannot be addressed by our numerical simulation method, but requires theoretical analyses.

As a computational modelling study, our work necessarily suffers from various limitations. For example, while the use of simple conductance based point neurons [42] is sufficient for our network study, perhaps even necessary for its tractability [67], it also limits our work. Unlike in the brain where calcium is compartmentalised in neurons [68], a single compartment point neuron model only allows one value of $[Ca^{2+}]$ for all neurites in a neuron. Thus, each of the neurons in our model can only either sprout or retract a type of neurite at a point in time. This is not the case in biology where different parts of the neuron can undergo structural changes independently of each other. The growth regimes suggested in our work must be understood to address the net formation or removal of neurites in neurons only. Furthermore, since a simultaneous homeostatic regulation of different neuronal compartments would be

expected to have a larger stabilising effect on the overall activity of the neuron, a single compartment neuron model may also limit the homeostatic effect of the structural plasticity mechanism. Point neurons also lack morphology, and our model is therefore unable to explicitly include the directional formation or removal of synapses. Axonal and dendritic arbours are not explicitly modelled in the MSP and the directional turnover of synapses that represents axonal sprouting emerges merely from the numbers of connecting partner neurites. Additionally, while it was enough for neurons in our model to be distributed in a two dimensional grid to include a spatial component, this is clearly not true for the brain. Thus, while our model provides a simplified high level view, the investigation of our proposed activity dependent growth rules in more detailed models is an important avenue for future research.

Finally, this work, and computational modelling of structural plasticity in general, are limited by the lack of supporting simulation tools and high computational costs. Most current simulators are designed for network modelling where synaptic connectivity remains constant. Even the NEST simulator [69], where the internal data structures are sufficiently flexible to allow for modification of synapses during simulation, currently includes a limited implementation of the MSP algorithm [70]. To incorporate the missing pieces—spatial information and different network connectivity modification strategies, for example—we were required to repeatedly pause simulations to make connectivity updates. This is far less efficient than NEST handling these changes in connectivity internally during continuous simulation runs and added a large overhead to the computational costs of our simulations. As a result, whereas the peripheral lesion experiments that are the foundation of this model are carried out over periods of months, only ˜20,000 s of simulation time could be simulated in 7 days of computing time on a computing cluster. Thus, the structural plasticity processes were considerably sped up in the model, and it is currently intractable to simulate them at biologically realistic time scales. The high computational costs associated with the model also prevent an exhaustive exploration of the multi-dimensional parameter space associated with the model. It follows that though the model makes predictions on the neurite growth in response to neuronal activity, we do not consider the presented growth curves to be optimally tuned and are unable to present a complete analysis of the parameter state space. The development of companion tools for modelling structural plasticity is however, gradually gaining traction [71] with discussions to allow NEST to communicate with stand alone structural plasticity tools via interfaces such as Connection Set Algebra [72] ongoing. The use of other computing technologies, such as Graphics Processing Units (GPUs) [73] and neuromorphic hardware [74, 75], for efficient simulation of structural plasticity remains an open research field.

In conclusion, we present a new general model of peripheral lesioning and repair in simplified cortical spiking networks with biologically realistic AI activity that provides several experimentally testable predictions.

## Methods

We build on and extend the Model of Structural Plasticity (MSP) [33] framework to model the activity dependent dynamics of synaptic elements. To honour our commitment to Open Science [76], we only made use of Free/Open source software for our work. We developed our new model using the NEST neural simulator [69, 77]. NEST includes an early, partial implementation of the MSP [70]. It does not, for example, currently take spatial information into account while making connectivity updates. More importantly, at this time, the design of the C++ code-base also does not provide access to the lower level rules governing updates in connectivity via the Python API. Making modifications to these to execute new structural plasticity connectivity rules, therefore, requires non-trivial changes to the NEST kernel. Given that work

is on-going to modularise the implementation of structural plasticity in NEST such that the computation of changes in connectivity will be left to stand-alone tools that will communicate them to the simulator using interfaces such as the Connection Set Algebra [72] (private communications with the NEST development team), we resorted to disabling connectivity updates in NEST. Instead, we generate connectivity based on our new hypotheses using native Python methods, and use the methods available in PyNEST to modify them in simulations. Our modified version of the NEST source code, based on the NEST 2.18.0 release [69], is available in our fork of the simulator available in a public repository at https://github.com/sanjayankur31/nest-simulator/tree/Sinha2020-str-p. The Vogels, Sprekeler Spike Timing Dependent Plasticity (STDP) model was contributed to the NEST simulator in version 2.12.0 [78]. Simulations without structural plasticity can, therefore, be run on any of the newer releases.

Simulations made use of the the University of Hertfordshire high performance computing cluster using computing 128 nodes. On this platform, the simulation time of ˜20,000 s (˜5 h) required 7 d of computing time. The peripheral lesion experiments that the model is based on, however, observe repair over a period of months. Thus, the structural plasticity mechanism is considerably fast forwarded in this model and does not run at biologically realistic time scales.

## Neuron model

Neurons are modelled as leaky integrate and fire conductance based point neurons with exponential conductances [42], the membrane potentials of which are governed by:

$$C\frac{dV}{dt} = -g_L(V - E_L) - g_{exc}(V - E_{exc}) - g_{inh}(V - E_{inh}) + I_e \tag{8}$$

where $C$ is the membrane capacitance, $V$ is the membrane potential, $g_L$ is the leak conductance, $g_{exc}$ is the excitatory conductance, $g_{inh}$ is the inhibitory conductance, $E_L$ is the leak reversal potential, $E_{exc}$ is the excitatory reversal potential, $E_{inh}$ is the inhibitory reversal potential, and $I_e$ is an external input current. Incoming spikes induce a post-synaptic change of conductance that is modelled by an exponential waveform following the equation:

$$g(t) = \bar{g}\exp\left(-\frac{t - t_s}{\tau_g}\right) \tag{9}$$

where $\tau_g$ is the decay time constant and $\bar{g}$ is the maximum conductance as the result of a spike at time $t_s$. Table 3 enumerates the constants related to the neuron model.

As in MSP, each neuron possesses sets of both pre- and post-synaptic synaptic elements, the total numbers of which are represented by ($z_{pre}$) and ($z_{post}$) respectively. The rate of change of each type of synaptic element, ($dz/dt$), is modelled as a Gaussian function of the neuron's Calcium concentration ($[Ca^{2+}]$) (Eqs (1) to (3)). Given that ($[Ca^{2+}] > 0$), ($dz/dt$) is bound as:

$$\begin{aligned}\min\left(\frac{dz}{dt}\right) &= -v\omega \quad \text{for} \quad ([Ca^{2+}] \to \infty) \\ \max\left(\frac{dz}{dt}\right) &= v(2 - \omega) \quad \text{for} \quad \left([Ca^{2+}] = \left(\frac{\eta + \epsilon}{2}\right)\right)\end{aligned} \tag{10}$$

If, based on its activity, a neuron has more synaptic elements of a particular type ($z$) than are currently engaged in synapses ($z_{connected}$), the free elements ($z_{free}$) can participate in the formation of new synapses at the next connectivity update step:

$$z_{free} = \lfloor(z - z_{connected})\rfloor \tag{11}$$

**Table 3. Neuronal parameters.**

| Parameter | Symbol | Value |
|---|---|---|
| LIF parameters | | |
| Refractory period | $t_{ref}$ | 5 ms |
| Reset potential | $V_{reset}$ | −60 mV |
| Threshold potential | $V_{th}$ | −50 mV |
| Capacitance | $C$ | 200 pF |
| Leak conductance | $g_L$ | 10 nS |
| Leak reversal potential | $E_L$ | −60 mV |
| Inhibitory reversal potential | $E_{inh}$ | −80 mV |
| Excitatory reversal potential | $E_{exc}$ | 0 mV |
| Excitatory time constant | $\tau_{exc}$ | 5 ms |
| Inhibitory time constant | $\tau_{inh}$ | 10 ms |
| $[Ca^{2+}]$ increase per spike | $\beta$ | 0.1 |
| $[Ca^{2+}]$ decay time constant | $\tau_{[Ca^{2+}]}$ | 50 s |
| External inputs | | |
| Poisson spike input to all neurons | $r_{ext}$ | 10 Hz |
| External projections to E neurons | $g_{ext}^E$ | 8 nS |
| External projections to I neurons | $g_{ext}^I$ | 12 nS |

However, if they remain unconnected, they decay at each integration time step with a constant rate $\tau_{free}$:

$$z_{free} = \lfloor (z_{free} - (\tau_{free} z_{free})) \rfloor \tag{12}$$

On the other hand, a neuron will lose $z_{loss}$ synaptic connections if the number of a synaptic element type calculated by the growth rules ($z$) is less than the number of connected synaptic elements of the same type ($z_{connected}$):

$$z_{loss} = \lfloor (z_{connected} - z) \rfloor \tag{13}$$

Table 4 lists the parameters governing the growth rules for all neurites.

## Network simulations

Our network model is derived from the cortical network model proposed by Vogels et al. [39] that is balanced by inhibitory homeostatic STDP. Like the cortex, this network model is characterised by low frequency Asynchronous Irregular (AI) [39, 40] firing of neurons. Additionally, this network model has also been demonstrated to store attractor-less associative memories for later recall. The simulation is divided into multiple phases, as shown in Fig 14. These are documented in the following sections in detail.

**Initial network structure.** We simulate a network of $N_E$ excitatory and $N_I$ inhibitory neurons ($N_E/N_I = 4$). Excitatory neurons are distributed in a two-dimensional rectangular plane such that the distance between two adjacent excitatory neurons is $(\mu_d^E \pm \sigma_d^E)$μm. Inhibitory neurons are scattered such that they are evenly dispersed among the excitatory neurons such that the mean distance between adjacent inhibitory neurons is $(\mu_d^I \pm \sigma_d^I)$μm. The rectangular plane is wrapped around as a toroid to prevent any edge effects from affecting the simulation. Table 5 summarises the parameters used to arrange the neurons.

At ($t = 0$ in Fig 14), neurons in the network are connected such that the network has a sparsity of $p$. For each neuron, $n_{out}$ targets are chosen from the complete set of possible post-

**Table 4. Growth rule parameters.**

| Parameter | Symbol | Value |
|---|---|---|
| **Optimal [$Ca^{2+}$]** | $\psi$ | |
| *Excitatory neurons* | | |
| Scaling factor: pre-synaptic structures ($z_{pre}^{E}$) | $v_{pre}^{E}$ | $15 \times 10^{-4}$ per $dt$ |
| Vertical shift | $\omega_{pre}^{E}$ | $1 \times 10^{-2}$ |
| X-axis parameters | $(\eta_{pre}^{E}, \epsilon_{pre}^{E})$ | $(\psi, 1.75 \times \psi)$ |
| Decay rate | $\tau_{pre,free}^{E}$ | 0.01 |
| Scaling factor: excitatory post-synaptic structures ($z_{post,E}^{E}$) | $v_{post,E}^{E}$ | $3 \times 10^{-5}$ per $dt$ |
| Vertical shift | $\omega_{post,E}^{E}$ | $4 \times 10^{-1}$ |
| X-axis parameters | $(\eta_{post,E}^{E}, \epsilon_{post,E}^{E})$ | $(0.25 \times \psi, \psi)$ |
| Decay rate | $\tau_{post,E,free}^{E}$ | 0.01 |
| Scaling factor: inhibitory post-synaptic structures ($z_{post,I}^{E}$) | $v_{post,I}^{E}$ | $3 \times 10^{-4}$ per $dt$ |
| Vertical shift | $\omega_{post,I}^{E}$ | $4 \times 10^{-2}$ |
| X-axis parameters | $(\eta_{post,I}^{E}, \epsilon_{post,I}^{E})$ | $(\psi, 3.5 \times \psi)$ |
| Decay rate | $\tau_{post,I,free}^{E}$ | 0.01 |
| *Inhibitory neurons* | | |
| Scaling factor: pre-synaptic structures ($z_{pre}^{I}$) | $v_{pre}^{I}$ | $3 \times 10^{-2}$ per $dt$ |
| Vertical shift | $\omega_{pre}^{I}$ | $4 \times 10^{-4}$ |
| X-axis parameters | $(\eta_{pre}^{I}, \epsilon_{pre}^{I})$ | $(0.25 \times \psi, \psi)$ |
| Decay rate | $\tau_{pre,free}^{I}$ | 0.01 |
| Scaling factor: excitatory post-synaptic structures ($z_{post,E}^{I}$) | $v_{post,E}^{I}$ | $3 \times 10^{-5}$ per $dt$ |
| Vertical shift | $\omega_{post,E}^{I}$ | $4 \times 10^{-1}$ |
| X-axis parameters | $(\eta_{post,E}^{I}, \epsilon_{post,E}^{I})$ | $(0.25 \times \psi, \psi)$ |
| Decay rate | $\tau_{post,E,free}^{I}$ | 0.01 |
| Scaling factor: inhibitory post-synaptic structures ($z_{post,I}^{I}$) | $v_{post,I}^{I}$ | $3 \times 10^{-5}$ per $dt$ |
| Vertical shift | $\omega_{post,I}^{I}$ | $4 \times 10^{-1}$ |
| X-axis parameters | $(\eta_{post,I}^{I}, \epsilon_{post,I}^{I})$ | $(\psi, 3.5 \times \psi)$ |
| Decay rate | $\tau_{post,I,free}^{I}$ | 0.01 |

synaptic neurons in a distance dependence manner as summarised in previous sections. Initially, static synapses in the network (II, IE, EI) are initialised to their mean conductances. The plastic (IE) synapses are subject to the homeostatic inhibitory synaptic plasticity mediated STDP rule proposed by Vogels, Sprekeler et al. [39] and are initialised to zero conductances.

External input to each neuron is modelled as an independent Poisson spike train with a mean firing rate $r_{ext}$. These spike trains project on to excitatory and inhibitory neurons via static excitatory synapses with conductances $g_{ext}^{E}$ and $g_{ext}^{I}$ respectively. Fig 1A shows the various sets of synapses in the network.

**Initial stabilisation to physiological state.** The simulation is then started and the network permitted to stabilise to its balanced AI state until ($t = t_2$ in Fig 14). Formally, the AI state is defined by Vogels, Sprekeler et al. [39] as:

$$(\text{ISI CV} > 1) \wedge (\sigma_{rate} < 5 \text{ Hz}) \tag{14}$$

where the ISI CV is the mean coefficient of variation of the inter-spike intervals (ISI) of

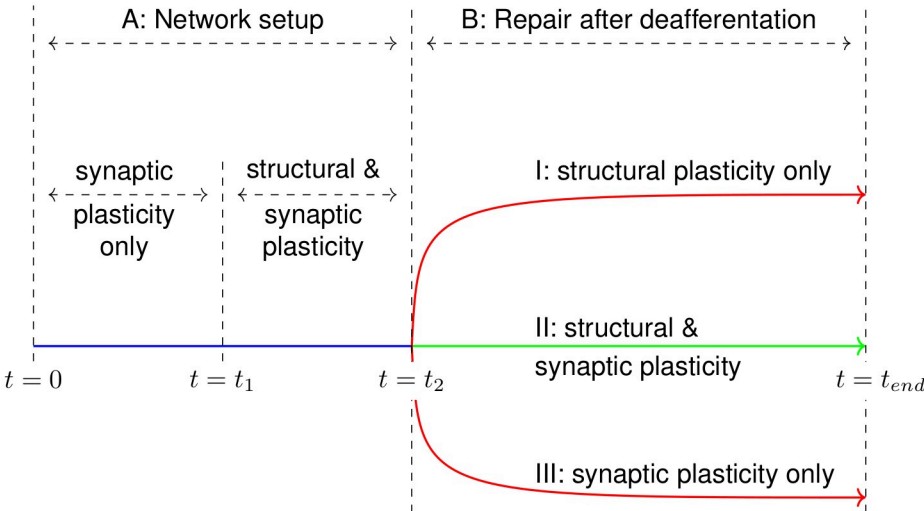

**Fig 14. The simulation runs in 2 phases.** Initially, the set-up phase ($0\,s < t < t_2$) is run to set the network up to the balanced AI state. At ($t = t_2$), a subset of the neuronal population is deafferented to simulate a peripheral lesion and the network is allowed to organise under the action of homeostatic mechanisms until the end of the simulation at ($t = t_{end}$). Each homeostatic mechanism can be enabled in a subset of neurons to analyse its effects on the network after deafferentation.

**Table 5. Network simulation parameters.**

| Parameter | Symbol | Value |
|---|---|---|
| Simulation parameters | | |
| Integration time step | $dt$ | 0.1 s |
| Structural plasticity update interval | | 1 s |
| Network parameters | | |
| Number of E neurons | $N_E$ | 8000 |
| Number of I neurons | $N_I$ | 2000 |
| Dimension of 2D E neuron lattice | | $100 \times 80$ |
| Dimension of 2D I neuron lattice | | $50 \times 40$ |
| Mean distance between E neurons | $\mu_d^E$ | 150 µm |
| STD distance between E neurons | $\sigma_d^E$ | 15 µm |
| Mean distance between I neurons | $\mu_d^I$ | 300 µm |
| STD distance between I neurons | $\sigma_d^I$ | 15 µm |
| Neurons in LPZ C | | 2.5% |
| Neurons in LPZ B | | 2.5% |
| Neurons in P LPZ | | 5% |
| Remaining neurons | | 90% |
| Initial network sparsity | $p$ | 0.02 |
| Initial out-degree | $n_{out}$ | $p \times$ total possible targets |
| Simulation stages | | |
| Synaptic plasticity only | | 1500 s |
| Synaptic and structural plasticity | | 500 s |
| Network deafferented at | | 2000 s |

neurons, and $\sigma_{rate}$ is the standard deviation of the population firing rate. We continue to use this formulation in this work.

Additionally, we also use the averaged pairwise cross-correlation between neurons in the network as an additional measure of synchrony in the network [43]. A population with a averaged pairwise cross-correlation of less than 0.1 is generally considered to be firing asynchronously. The Elephant analysis toolkit (version 0.10.0) [79] was used to make the calculation with a bin size of 5 ms (we also tested with bin sizes of 2 ms, 10 ms, 20 ms, and 50 ms and received similar results). The toolkit calculates the cross-correlation between each pair of a given set of spike trains. For populations of fewer than 800 neurons, we considered all neurons in the population. For populations of more than 800 neurons, we considered either 10% of the total number of neurons or 800—whichever was greater.

The initial stabilisation phase consists of two simulation regimes. Initially, only inhibitory synaptic plasticity is activated to stabilise the network ($t < t_1$ in Fig 14). As this state ($t = t_2$ in Fig 14) is considered the normal physiological state of our network model, the network parameters obtained at this point are set as the steady state parameters of neurons and synapses in the network. The optimal activity of each neuron, $\psi$, is set to the activity achieved by the neuron at this point, and its growth curves are initialised in relation to it. Since neurons in the AI network have similar but not identical activity, it follows that their optimal activities are also similar but not necessarily identical. This ensures that the structural plasticity mechanism attempts to stabilise all neurons to the activities they achieved when the network is balanced by the inhibitory STDP to its normal AI state. The mean conductance for new IE synapses is also set as the mean conductance of the IE synapses obtained at this stage.

Our implementation of structural plasticity is then activated in the network at this point ($t = t_1$ in Fig 14) to verify that the network continues to remain in its balanced AI state in the presence of both homeostatic mechanisms.

**Simulation of peripheral lesion.** Next at ($t = t_2$ in Fig 14), the external Poisson spike train inputs are disconnected from excitatory and inhibitory neurons that fall in the Lesion Projection Zone (LPZ) to simulate a peripheral lesion in the network.

**Network reorganisation.** The deafferented network is permitted to reorganise itself under the action of the active homeostatic mechanisms until the end of the simulation ($t = t_{end}$ in Fig 14). By selectively activating the two homeostatic mechanisms in different simulation runs, we were also able to investigate their effects on the network in isolation.

**Structural plasticity mediated connectivity updates.** All synapses in the network, except the connections that project the external stimulus on to the neuronal population, are subject to structural plasticity (Fig 1A).

Free excitatory pre-synaptic and excitatory post-synaptic elements can combine to form excitatory synapses (EE, EI). Analogously, inhibitory pre-synaptic and inhibitory post-synaptic elements can plug together to form inhibitory synapses (II, IE). The set of possible partners for a neuron, therefore, comprises of all other neurons in the network that have free synaptic elements of the required type. From this set, $z_{free}$ partners are chosen based on a probability of formation, $p_{form}$, which is a Gaussian function of the distance between the pair, $d$:

$$p_{form} = \hat{p}\exp^{-(d/(w\mu_d^E))^2} \tag{15}$$

Here, $\hat{p} \in \{\hat{p}_E, \hat{p}_I\}$ is the maximum probability, $\mu_d^E$ is the mean distance between two adjacent excitatory neurons, and $w \in \{w_E, w_I\}$ is a multiplier that controls the spatial extent of new synaptic connections.

Investigations indicate that lateral connections in the primary visual cortex are organised in a "Mexican hat" pattern. While experimental work does support the presence of the "Mexican

hat" pattern [80, 81], anatomical research suggests that inhibitory connections are more local-ised than excitatory ones, contradicting the traditional use of shorter excitatory and longer inhibitory connections in computer models [82]. Analysis of the local cortical circuit of the primary visual cortex suggests that the "Mexican hat" pattern can either be generated by nar-row but fast inhibition, or broad and slower inhibition that may be provided by longer axons of GABAergic basket cells [83, 84]. Investigations into the maintenance of the "Mexican hat" pattern are beyond the scope of this study. We therefore, limit ourselves to the traditional model of longer inhibitory connections and shorter local excitatory connections in this work by using a larger multiplier for inhibitory synapses, $w_I$, than for excitatory synapses, $w_E$, ($w_E <$ $w_I$).

New synapses that are added to the network are initialised with conductances similar to that of existing synapses in the balanced network. Their conductance values are taken from a Gaussian distribution centred at the mean conductance for that synapse type. Since new syn-apses can, therefore, be weaker or stronger than existing ones, this prevents the same set of synapses from being modified in each connectivity update.

In spite of them being plastic, the same method is also used for IE synapses. IE synapses are initialised with zero conductances at the start of the simulation and modify their strengths based on STDP [39]. When the network has achieved the balanced AI state, these conduc-tances also settle at higher values. If new IE synapses formed after this point by structural plas-ticity were to be initialised to zero conductances, they would most likely be selected for deletion repeatedly as the weakest ones. STDP does not modulate inactive synapses either—synapses between pairs of neurons that have both been rendered inactive by deafferentation will not be weakened, and may not be lost. Therefore, to ensure the turnover of a diverse set of IE synapses also, new connections of this type are supplied with conductances similar to that of existing stable IE synapses in the balanced network.

Experimental evidence suggests that the stability of synapses is proportional to their efficacy [13, 85]. Taking this into account, we calculate the probability of deletion of a synapse, $p_{del}$, as a function of its conductance $g$:

$$p_{del} = \exp^{-\left(\frac{g}{(2g_{th})}\right)^2} \tag{16}$$

Here, $g_{th}$ is a threshold conductance value calculated during the simulation, synapses stronger than which are considered immune to activity dependent changes in stability. They are removed from the list of options from which $z_{loss}$ synapses are selected for deletion and are therefore, not considered for deletion at all.

For simplicity, for static excitatory synapses that all have similar conductances (EI, EE), we do not use this method of deletion. Instead, for these, $z_{loss}$ connections are randomly selected for deletion from the set of available candidates. While II synapses are also static, the deletion of an inhibitory synapse by the loss of an inhibitory post-synaptic element can occur by the removal of either an IE or an II synapse. Therefore, to permit competition between II and IE synapses for removal, we apply weight based deletion to both these synapse sets.

The numbers of synaptic elements are updated at every simulator integration time step internally in NEST. Connectivity updates to the network, however, require updates to internal NEST data structures and can only be made when the simulation is paused and incur consider-able computational costs. Given that the synaptic plasticity mechanism acts at the time scale of milliseconds ($\tau_{STDP}$ = 20 ms), we make connectivity updates at 1 s intervals to keep structural plasticity updates faster than that in biology, but still sufficiently slower than the synaptic plas-ticity mechanism. Gathering data on conductances, connectivity, and neuronal variables like

**Table 6. Synapse parameters.**

| Parameter | Symbol | Value |
|---|---|---|
| Unit conductance | $\bar{g}$ | $(0.5 \pm 0.1)$ nS |
| EE synapse conductance | $g_{EE}$ | $\bar{g}$ |
| EI synapse conductance | $g_{EI}$ | $\bar{g}$ |
| II synapse conductance | $g_{II}$ | $10\bar{g}$ |
| IE synapse conductance | $g_{IE}$ | Vogels, Sprekeler STDP |
| STDP rule time constant | $\tau_{STDP}$ | 20 ms |
| Target constant | $\alpha_{STDP}$ | 0.12 |
| STDP learning rate | $\eta_{STDP}$ | 0.05 |
| Width multiplier: excitatory synapses | $w_E$ | 8 |
| Width multiplier: inhibitory synapses | $w_I$ | 24 |
| Maximum probability of formation: excitatory synapses | $\hat{p}_E$ | 0.8 |
| Maximum probability of formation: inhibitory synapses | $\hat{p}_I$ | 0.3 |
| Conductance threshold for deletion: inhibitory synapses | $g_{th}$ | |

$[Ca^{2+}]$ also require explicit NEST function calls while the simulation is paused. Therefore, we also limit dumping the required data to files to regular intervals. Table 6 summarises the various synaptic parameters used in the simulation.

## Single cell simulations

We also studied the effects of our structural plasticity hypotheses in individual neurons using single neuron simulations. Fig 10A shows a schematic of our single neuron simulations.

The neuron is initialised to a steady state where it exhibits an in-degree similar to neurons in the network simulations when in their AI state. To do so, a constant baseline input current $I_{ext}$ is supplied to the neuron to provide it with activity. The $[Ca^{2+}]$ obtained by the neuron at this time is assumed as its optimal level, $\psi$. Using identical values of $\eta$ and $\epsilon$ but different $\nu$ values for excitatory and inhibitory post-synaptic elements ($\nu_{post}^E = 4\nu_{post}^I$ to mimic the initial in-degree of neurons in our network simulations), and an input current that deviates the activity of the neuron off its optimal level ($< I_{ext}$), the neuron is made to sprout $z_{post}^E, z_{post}^I$ excitatory and inhibitory post-synaptic elements respectively ($z_{post}^E = 4z_{post}^I$). At this stage, the neuron has been initialised to resemble one in network simulations in its balanced state before deafferentation. The current input is returned to its baseline value, thus returning the $[Ca^{2+}]$ to its optimal value, $\psi$.

Next, the growth curves for the neuron are restored as per our activity dependent structural plasticity hypotheses to verify that the neuron does not undergo any structural changes at its optimal activity level. The external current input to the neuron is then modulated sinusoidally to fluctuate the neuron's $[Ca^{2+}]$ (Fig 10B), and resultant changes in the numbers of its post-synaptic elements are recorded. By assuming that each dendritic element receives inputs via conductances as observed in network simulations ($g_{EE}, g_{IE}$), the net input to the neuron that results in its activity can be approximated as:

$$g_{net} = z_{post}^E g_{EE} - z_{post}^I g_{IE} \tag{17}$$

As the neuron modifies its neurites, the change in excitatory and inhibitory input conductance received as a result is calculated (Fig 10C).

## Acknowledgments

We are grateful to Benjamin Torben-Nielsen for fruitful discussions and feedback on the work. We are also most grateful to the NEST development team, in particular to Sandra Diaz-Pier, for discussions and assistance with the modelling of structural plasticity in the NEST simulator.

## Author Contributions

**Conceptualization:** Ankur Sinha, Christoph Metzner, Neil Davey, Roderick Adams, Volker Steuber.

**Formal analysis:** Ankur Sinha, Christoph Metzner, Neil Davey, Michael Schmuker, Volker Steuber.

**Investigation:** Ankur Sinha, Christoph Metzner.

**Methodology:** Ankur Sinha, Christoph Metzner, Neil Davey, Roderick Adams, Michael Schmuker, Volker Steuber.

**Project administration:** Ankur Sinha, Christoph Metzner, Roderick Adams, Michael Schmuker, Volker Steuber.

**Resources:** Ankur Sinha, Volker Steuber.

**Software:** Ankur Sinha.

**Supervision:** Christoph Metzner, Neil Davey, Roderick Adams, Michael Schmuker, Volker Steuber.

**Visualization:** Ankur Sinha, Christoph Metzner, Roderick Adams, Volker Steuber.

**Writing – original draft:** Ankur Sinha.

**Writing – review & editing:** Christoph Metzner, Neil Davey, Roderick Adams, Michael Schmuker, Volker Steuber.

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
