## [Decision Letter · Decision Letter 0]

29 Jul 2020

Dear Mr Sinha,

Thank you very much for submitting your manuscript "Growth Rules for the Repair of Asynchronous Irregular Neuronal Networks after Peripheral Lesions" for consideration at PLOS Computational Biology.

As with all papers reviewed by the journal, your manuscript was reviewed by members of the editorial board and by several independent reviewers. In light of the reviews (below this email), we would like to invite the resubmission of a significantly-revised version that takes into account the reviewers' comments.

In particular, there are a number of claims in the current version of the manuscript that are insufficiently supported by the findings presented.  As a major example of this, Reviewer 2 points out that that there is neither sufficient evidence that the 2013 study was not in the AI regime, nor that the networks of the current study are in that regime. Beyond this, the novelty with respect to previous studies needs to be core carefully elaborated, and the reviews identify a number of areas in which the clarity of the paper could be improved.

We cannot make any decision about publication until we have seen the revised manuscript and your response to the reviewers' comments. Your revised manuscript is also likely to be sent to reviewers for further evaluation.

Sincerely,

Abigail Morrison

Associate Editor

PLOS Computational Biology

Lyle Graham

Deputy Editor

PLOS Computational Biology

Reviewer's Responses to Questions

**Comments to the Authors:**

Reviewer #1: Understanding the mechanisms that enable the brain to develop and then to recover from insults is a very important, but still understudied, topic in neuroscience. Computational modelling has a major role to play in such research by allowing precise investigation of the effects of different “rules” that determine plasticity, both synaptic and structural, in neural networks. This manuscript presents a study that adds significant new aspects to the earlier work of Butz and van Ooyen (MSP model) and provides some interesting predictions that can be the subject of future experimental investigation.

The manuscript is generally well written and presented, but could do with clarification of several points:

1. It is unclear whether all the main simulation results presented include all forms of structural plasticity: this would seem to be the case eg. Fig 3 caption just refers to “our structural plasticity mechanism”. But the Results section then discusses the post-synaptic changes and the pre-synaptic changes separately, which led me to think you trialled them one at a time (which I do not think you did). This should be clarified in the text.

a. Also, combining tables 1 and 2 would allow for easier comparison of the pre- and post-synaptic element rules

2. Reproducing the time course of recovery is highlighted a number of times (eg line 258 in the Discussion) but it is not made precise what features of the time course are apparent from the simulations.

a. Is it mostly the distinction between activity seen just post the lesion time and that arrived at in the steady-state?

b. How does simulation time relate to real time in the nervous system: are you using realistic time scales?

c. What does experimental data indicate about the time course of axon outgrowth and synapse formation or loss?

d. What determines the time course in the model? It would seem that the major determinant is the change in [Ca] level as determined by its decay time constant, and this is applied uniformly to all forms of structural plasticity. Does experimental data indicate different time courses for axonal outgrowth versus synapse formation / loss and could this be accounted for through the use of different time constants for the separate plasticity processes?

3. Clarify in the text that the peri-LPZ is not part of the deafferented zone, but rather is the area referred to when talking about neurons just outside the LPZ. Or if I am mistaken, then definitely clarify what neurons are outside the LPZ and where we can see the increase in activity in these following a lesion (Figures 3 and 4 supposedly illustrate this but only show LPZ-C and peri-LPZ).

4. The plots in figures 3 and 4 seem a little counter-intuitive: in figure 3, the firing rates remain somewhat variable in the LPZ throughout, yet there is a marked reduction in variability in the peri-LPZ; whereas in figure 4, the CV of the ISI drops in the LPZ (and how is this calculate if there is no activity in the LPZ following the lesion?), but remains somewhat static in the peri-LPZ?

a. Also, fig 3A and 4B would both be improved by a clear indication of the neurons inside the LPZ eg circles in 3A and a line covering all the panels in 4B (not just at the y-axis).

5. Can the increase in firing in the peri-LPZ following the lesion be quantified eg as a percentage of the pre-lesion firing level? It would seem to be more significant in the inhibitory neurons (fig 3C), but maybe it is not, if quantified.

6. How, exactly, were the growth rules tuned (eg line 132)? Was it purely done qualitatively on the basis of most neurons returning to a pre-lesion mean firing rate? Was the level of AI quantified in some way and also used as a measure (at least for neurons outside the LPZ) to tune against?

7. How significant is the size of the LPZ (relative to the network size) for the time course of recovery and the emergence of the synchronous activity pattern seen in the LPZ?

8. The Discussion neglects to talk about (lines 288 onwards) the outgrowth of inhibitory axons from the LPZ, experimental evidence for which was cited in the introduction [6], and forms an important outcome of the simulations.

Reviewer #2: The manuscript addresses possible mechanisms of repair in neuronal networks that are partially deprived of their input, for example as a consequence of peripheral lesions. The scientific scope of the work is very similar to a paper published in PLOS Computational Biology many years ago (Butz & van Ooyen, 2013). The core concept of the solution proposed in the new manuscript – homeostatic structural plasticity – was in fact brought up in several papers by Arjen van Ooyen and colleagues around the year 2009. The claim of the new manuscript is that the original idea does not work any more in the biologically more realistic setting of asynchronous-irregular (AI) network activity. Imposing additional functional plasticity to all inhibitory synapses in the network, however, is found to mitigate these problems, and successful repair could be demonstrated in numerical simulations.

In view of the strong overlap in concepts and results of the new submission and a previous publication, the question arises, whether the novelty represented by the submitted manuscript deserves a separate publication of it at all. For that reason, I will discuss strengths and weaknesses of the new manuscript in comparison to the previous publication.

(1) In their introduction, the authors of the new study make the following statement: “Since the peripheral lesion model proposed by Butz and van Ooyen [33] was not based on a balanced cortical network model with biologically realistic AI activity, their hypothesised growth rules did not elicit repair in our simulations.” This claim provides justification to take up the issue again and propose a new solution to it. The logic of this statement, however, is problematic, as no analysis of sufficient generality was performed in the submitted manuscript. Whereas the authors of the original study used the Izhikevich neuron model, the new study employs an integrate-and-fire neuron model with conductance-based synapses. In both cases, 80% of all neurons were excitatory, and the remaining 20% inhibitory. The original study considered networks comprising 400 neurons in total, whereas the new study seems to have performed simulations of a 25-fold larger network, following Vogels et al. (2011). The actual numbers used are not revealed in the manuscript, however.

What then is the basis of the claim that the 2013 paper did not consider networks in the AI state? This issue was briefly addressed in Fig. 12 of the original publication, which displays only population activity traces, no spike trains. Although no formal analysis was performed, the visual appearance of the traces shown indeed suggests AI-like activity. Interestingly, the authors of the new paper do not provide any formal analysis of this issue, neither concerning their own networks, nor the networks of the old paper. The activity states of their simulated networks are not characterized quantitatively, and the spike trains shown in Fig. 4 of the submitted manuscript do not allow any conclusions either, they do not even have a proper time axis. Therefore, a convincing quantitative underpinning of the above-mentioned claim is inevitable.

(2) The authors of the new study claim that the hypothesized growth rules did not elicit repair in their simulations. This statement is also very problematic. Maybe the authors just have not tried hard enough to get it to work. Many parameters are different compared to the original setting, so it would not be surprising if also some of the parameters of the plasticity rule need adjustment. The only acceptable argument why it cannot work would be based on a mathematical analysis of the situation, but no such analysis is provided in the manuscript. Instead, the authors suggest an entirely new component of the model (inhibitory STDP) to make network repair possible. This scenario could be accepted as one possible (“sufficient”) solution to the problem. In their abstract, however, the authors make the claim that their solution is the only one possible (“necessary”). They claim “Lastly, we observe that our proposed model of homeostatic structural plasticity and the inhibitory synaptic plasticity mechanism that also balances our AI network are both necessary for successful rewiring of the network.” No result presented in the manuscript can support this claim. This issue needs clarification.

(3) The new solution proposed in the manuscript involves two different types of synaptic plasticity, homeostatic structural plasticity and inhibitory spike-timing dependent plasticity. Whereas the latter is relatively fast with time scales in the range of tens of milliseconds, the former is rather slow, operating on time scales that are several orders of magnitude larger. But how then can fast inhibitory plasticity compensate for the deficits of slow homeostatic plasticity? On page 9, second to last paragraph the authors state: “Simulations require the growth rates of inhibitory axonal elements to be high enough to stabilise the large number of hyperactive neurons outside the LPZ.” It remains unclear why inhibitory plasticity cannot compensate for this.

Generally, the new manuscript does not provide sufficient insight into the question how the two types of plasticity interact. For example, a simple tracking of inhibitory amplitudes in the course of time, together with structural changes that happen simultaneously, would shed light on this question. Such extra insight is absolutely necessary to justify a solution that is considerably more complicated than previous suggestions.

(4) The authors of the submitted manuscript discuss different options for the growth curves describing the homeostatic controller. However, they discuss only growth curves that arise from a translation of the Gaussian curve in x and y direction. What is the exact motivation of this somewhat restricted perspective? As the most important parameter is the slope of the curve in the set-point, other transformations (e.g. a stretching) of the growth curves actually seem more relevant. Why does one need a Gaussian shape for the axonal elements to begin with? Wouldn't the creation of axonal elements at a constant rate provide a simpler (better) solution? A more systematic account of these questions is necessary to go beyond the insight from previous studies.

Reference

Butz M, van Ooyen A

A Simple Rule for Dendritic Spine and Axonal Bouton Formation Can Account for Cortical Reorganization after Focal Retinal Lesions

PLoS Computational Biology 9(10): e1003259, 2013

https://doi.org/10.1371/journal.pcbi.1003259

Reviewer #3: A propery formatted version of this review has been uploaded as a PDF.

The present study proposes a simple yet qualitatively significant change to the MSP model, proposed by Butz et al, that provides a generic framework to describe post-lesion repair.

They apply this framework to peripheral lesioning of a balanced AI network aimed at modeling cortical activity.

In addition to the change in the MSP model, they also include I->E STDP plasticity and strength-dependent deletion of synapses to their simulation, then assess numerically how both plasticity mechanisms must work together to generate a partial recovery.

The rationale behind the evolution of post- and pre-synaptic structures for both collective and single-cell dynamics also seems clear and consistent, even though limited experimental and numerical evidence is provided as to why other possibilities should not be considered.

Required code to reproduce the result is provided using free open-source licenses.

I believe that this work represents an interesting contribution and a useful, logical advancement for studies on structural plasticity.

I would support its publication in PLOS CB provided one major question is answered and some restructuring is performed on the manuscript, including the addition of precision/discussions (notably a more in-depth comparison with previous studies).

The one major issue that I have with this study concerns the stability of the network activity with respect to the rules chosen for structural plasticity (SP).

Indeed, the authors only looked at the influence of structural plasticity for 500 s in the control state.

This seems too short compared to the typical timescales over which the changes are occurring during the recovery (several thousand seconds).

Furthermore, most growth curves are intrinsically unstable on their own, making the reason for the overall stability hard to understand, especially as it is not discussed at all (though, to be fair, the non-trivial case of inhibitory neurons was also ignored by Butz et al. in the original model [33]).

Besides this major issue, I feel that there is a need for more thorough justification of several modeling choices and their consequences:

A. Are the choices related to the growth curves really ensuring the stability of the activity?

B. Was the final increase in synchrony not expected given the chosen parameters?

C. Why was STDP limited to I->E and instead of (for instance) scaling of excitatory synapses?

D. Structural changes are very fast compared to the biological values mentioned in the MSP paper [33] (72 days over the whole timecourse) and the experimental studies that are cited [6, 9] (several days). The necessity of speeding up simulations is understandable (it is also argued in [33]) and it is probably sufficient for the timescales of neuronal activity/STDP and SP to occur on "sufficiently different" timescales to achieve "reasonable modeling". However that "sufficient difference" should certainly be discussed if not proven, especially given the fast growth of inhibitory axons in the current study.

More specific remarks and questions regarding precise parts of the manuscript are detailed below, they are usually related to points A and B.

English and structure

Overall the structure of the article should be revised to clarify the novelty and hypotheses underlying the study and the differences with previous studies.

In particular, care should be taken to avoid duplicates and verbosity in the main text, while increasing the amount of information present in the captions of the figures which is currently limited.

This issue is especially visible between the Results and Methods, where equations 1-3 and 10-13 are the same.

Various issues:

* Missing hyphens for compound adjectives.

* L280 should read "consistent" and not "coherent".

* Remove double citation of [3] in lines 285-286.

Scientific issues

Introduction

----------------

The statement

> Additionally, while providing salient testable predictions, the original MSP growth rules have specifically been developed for excitatory neurites only—they do not provide activity dependent growth rules for inhibitory neurites, nor do they reproduce the experimentally observed outgrowth of inhibitory axons from the LPZ.

is strange since the MSP model does include structural plasticity for inhibitory synapses.

See equations 6 and 8 of [33] https://journals.plos.org/ploscompbiol/article?id=10.1371/journal.pcbi.1003259 and the sentence

> We postulate Gaussian-shaped growth curves for the activity- dependent formation and deletion of every type of synaptic element, i.e. excitatory and inhibitory axonal elements A, excitatory dendritic elements Dex and inhibitory dendritic elements Din

Furthermore, though the associated outgrowth of inhibitory axons may not reproduce some observations, the absence of a real convergence in observations of inhibitory plasticity because of their great variability make this point rather weak (see latter comments in Discussion).

Results

----------

**A. Stability and growth curves**

Paragraph above line 73: dz/dt = 0 is not sufficient to provide stability, and the condition for stability depends on the neurons type and incoming/outgoing connections.

Generally, stability in the system is complex and not properly defined nor analyzed. This is clearly visible on Figure 3: where the 500 s window to assess the stability of the AI state seems too short given the typical evolution we see afterwards. I would expect the timescale of changes to be at least 2000 s and I do not think that the simulations shown can properly assess the network's stability with respect to SP.

Tables 1 and 2 are extremely unclear, notably regarding how these facts are verified in simulations (especially as "stability" is not defined). What does the absence of entry for repair in the middle column mean?

Lines 201-204: what does the following sentence mean by "stable state"?

> While a few other pre-synaptic growth curves did allow simulations to show an increase in activity in the LPZ and a loss of activity outside it, the networks in these simulations did not re-balance to a stable state.

Similarly, lines 209-210, what is meant by "stabilize" and "hyperactive"?

> simulations require the growth rates of inhibitory axonal elements to be high enough to stabilize the large number of hyperactive neurons outside the LPZ

What happens if the growth rate is not high enough?

Most of the growth curves chosen on Figure 5 are not stable in themselves but only when combined together or thanks to the network properties; this fact should be analyzed and discussed in more details. For instance, for excitatory neurons, the stable fixed-point of the activity is \\eta, the instability of the z^E_{pre} curve is prevented by the choice of a stable curve for z^E_{post}, and how the instability of z^I_{pre} might be stabilized by through network retroactions is non-trivial, especially given how \\psi is chosen (see remark in Methods).

The simple fact that this system may not be stable could, in itself, explain the results shown on Figure 13.

**B. Increased synchrony**

From the results in Butz et al. [33], one may suspect that this high growth rate and low \\eta^I_{pre} are the reason for the more synchronous behavior that is shown on Figure 4; this should be discussed.

Discussion

---------------

258-259: the qualitative changes might be recovered, but not the actual timecourse since the timescales are way shorter.

**A. Stability and growth curves**

Lines 284-286: If the study by Knott et al [3] finds a proportional increase in the number of inhibitory spines associated to increased stimulation, their observations explicitly states that their is also a significant increase in the number of excitatory spines, this fact goes against the proposed rule for excitatory spines.

Overall, evidence supporting any specific choice of growth curves seems quite limited, which is why (unless stronger numerical evidence is provided) some additional care should be taken, acknowledging that while the chosen set is reasonable, other combinations or parameters may also lead to similar if not more suitable results.

In particular, this considerably weakens the statement on the loss of the AI state after repair (see remark B below).

Lines 313-315: "Finally, our simulation results indicate that the suggested growth rules, while derived from network simulations, can contribute to the stability of activity in individual neurons (Fig. 10)."

This assertion and the experiment illustrated on Figure 10 are true under the hypothesis of a balanced network, with homogeneous synaptic weights, near its steady-state (which is in itself an interesting result).

Whether SP has the same effect in the situation studied in the manuscript (where the whole network goes away from its previous steady state, individual neurons have synapses with different weights, and the input received in the LPZ are a priori unbalanced and depend on the neuron's own activity and type) or in general is however not a trivial question.

The activity resulting from SP alone and the synchronization obtained in the "successful recoveries" already cast some doubts about this general stabilizing effect, which is why I think the underlying assumptions should be specified.

**B. Increased synchrony**

266-267: it seems to me that a switch from AI to synchronous would greatly affect the function of the network and may point out at limitations of the model rather than anything else, especially given that this result was expected based on the initial study by Butz et al. [33].

Until a more exhaustive numerical exploration (or analytical analysis) of the parameter space is performed, there is no reason to believe that a recovery maintaining the AI activity is impossible.

Methods

-----------

Lines 542-543: This sentence sounds like that the growth curves for excitatory and inhibitory spines are the same (\\eta_I = \\eta_E, \\epsilon_I = \\epsilon_E), which should not be the case according to the choices made.

The evolution shown on Figure 10 suggests that the correct growth curves (shown on Figure 5) were used; the sentence should therefore be clarified to state the difference between the growth curves, with \\eta_I = \\epsilon_E = \\psi.

One may actually wonder, though, whether the distinction between excitatory and inhibitory dendritic elements in the model makes sense (since, to the best of my knowledge, this distinction has no biological origin).

It is worth noting that, in the original model, this distinction had much weaker implications since the parameters for both types where identical.

Table 5 : \\nu entries should have units

**A. Stability and growth curves**

Lines 451-452: If I understand correctly, the value of \\psi for differs for each neuron in the network.

If so, the choice to assign a given \\psi to each neuron based on the activity of that neuron at a precise point in time, due to the STDP, seems rather arbitrary, especially since the STDP and the homeostatic plasticity have separate roots.

Wouldn't it make more sense to choose the same \\psi for all inhibitory neurons and another one for the excitatory neurons (e.g. the median or average values)?

I think a justification of this choice and a discussion of its consequences are necessary.

Implementation issues

1) Though the changes made to the NEST kernel are straightforward, they may quickly pose a significant challenge to most users that would want to reproduce the results (as installation methods and platform requirements change) or assess the impact of bugs found in NEST after the fork diverged from the main repository (which was already more than two years ago, in June 2018).

Furthermore, citation for NEST is "Jordan J, Mørk H, Vennemo SB, Terhorst D, Peyser A, Ippen T, et al. NEST 2.18.0; 2019. Available from: " ext-link-type="uri" xlink:type="simple">https://doi.org/10.5281/zenodo.2605422", which seems wrong if the code used indeed corresponds to the linked branch.

I would suggest to merge the NEST 2.20 release into the branch so that this issue is at least be postponed by a couple of years and so that users have an easier way to assess changes and bugs: they would be compared to a release rather than a random commit.

Similarly, for simulations that do not involve structural changes, this would tell users that they should be able to reproduce them using the 2.20 release.

2) It is unclear how the timestep used to update the structural plasticity (1 second in this study) may affect the results; I think that a brief investigation of its influence would be necessary, if only as supplementary information, and compared to the timestep used in Butz et al. (100 ms).

**Have all data underlying the figures and results presented in the manuscript been provided?**

Reviewer #1: Yes

Reviewer #2: **No: **There is no explicit mention of the size of the simulated networks.

Reviewer #3: Yes

PLOS authors have the option to publish the peer review history of their article (what does this mean?). If published, this will include your full peer review and any attached files.

Reviewer #1: **Yes: **Bruce Graham

Reviewer #2: No

Reviewer #3: **Yes: **Tanguy Fardet
---

## [Decision Letter · Decision Letter 1]

23 Jan 2021

Dear Mr Sinha,

Thank you very much for submitting your manuscript "Growth Rules for the Repair of Asynchronous Irregular Neuronal Networks after Peripheral Lesions" for consideration at PLOS Computational Biology.

As with all papers reviewed by the journal, your manuscript was reviewed by members of the editorial board and by several independent reviewers. In light of the reviews (below this email), we would like to invite the resubmission of a significantly-revised version that takes into account the reviewers' comments.

The majority of the remaining issues have a common thread, namely a disparity between what your data show and what conclusions can be drawn from them. As a clear-cut example of this, reviewer 2 points out that a numerical simulation showing repair when synaptic and structural plasticity are coupled, plus the statement that repair was not possible (in your hands, with your assumptions, with the parameter regimes you investigated) does not entail that the synaptic plasticity is a necessary condition. For this, one would have to have a theoretical analysis. However, I do not regard this as an insurmountable issue: the results are clearly interesting as they are! In most cases the comments can be handled by being more careful about what is concluded (especially being clear about the difference between a possible interpretation and a conclusion), and explicitly recognising the limits of a purely numerical approach. In other cases, for example, the question of AI metrics, it seems that plotting some additional metrics (in this case, a synchrony measure) will help to convince the reader that you have carefully excluded other, alternative interpretations of your results.

We cannot make any decision about publication until we have seen the revised manuscript and your response to the reviewers' comments. Your revised manuscript is also likely to be sent to reviewers for further evaluation.

Sincerely,

Abigail Morrison

Associate Editor

PLOS Computational Biology

Lyle Graham

Deputy Editor

PLOS Computational Biology

Reviewer's Responses to Questions

**Comments to the Authors:**

Reviewer #1: The authors have satisfactorily addressed all my comments on their original submission of this manuscript.

Reviewer #2: The review is uploaded as an attachment.

Reviewer #3: I thank the authors for their answers and modifications to the manuscript.

My comments and concerns have been appropriately answered.

Since the authors have also included two Figures in there response, though, I would suggest these Figures be included in the Supplementary Information and appropriately mentioned in the main text for completeness.

Other than this minor recommendation, I believe that the manuscript is now suitable for publication in PLOS CB.

**Have all data underlying the figures and results presented in the manuscript been provided?**

Reviewer #1: Yes

Reviewer #2: **No: **

Reviewer #3: Yes

PLOS authors have the option to publish the peer review history of their article (what does this mean?). If published, this will include your full peer review and any attached files.

Reviewer #1: No

Reviewer #2: No

Reviewer #3: **Yes: **Tanguy Fardet
---

## [Editor Report · Decision Letter 2]

23 Apr 2021

Dear Mr Sinha,

We are pleased to inform you that your manuscript 'Growth Rules for the Repair of Asynchronous Irregular Neuronal Networks after Peripheral Lesions' has been provisionally accepted for publication in PLOS Computational Biology.

***

Can I recommend that you carry out a careful proof-read at this point? I noticed that the sentence starting on l. 327 is incomplete, there may be others.

***

Best regards,

Abigail Morrison

Associate Editor

PLOS Computational Biology

Lyle Graham

Deputy Editor

PLOS Computational Biology

---

## [Editor Report · Acceptance letter]

24 May 2021

PCOMPBIOL-D-20-00647R2

Growth Rules for the Repair of Asynchronous Irregular Neuronal Networks after Peripheral Lesions

Dear Dr Sinha,

I am pleased to inform you that your manuscript has been formally accepted for publication in PLOS Computational Biology. Your manuscript is now with our production department and you will be notified of the publication date in due course.

With kind regards,

Katalin Szabo
